# The sizes of life

Eden W. Tekwa[1,2]*, Katrina A. Catalano[2], Anna L. Bazzicalupo[1], Mary I. O'Connor[1], Malin L. Pinsky[2]

**1** Department of Zoology, University of British Columbia, Vancouver, BC, Canada, **2** Department of Ecology, Evolution and Natural Resources, Rutgers University, New Brunswick, NJ, United States of America

* ewtekwa@gmail.com

**Data Availability Statement:** All data and code are available on https://github.com/EWTekwa/BodySizeBiomass.

**Funding:** The research was supported by the Tula Foundation (ET); the Gordon and Betty Moore Foundation (MP); MITACS (ET); the Coral Reef

## Abstract

Recent research has revealed the diversity and biomass of life across ecosystems, but how that biomass is distributed across body sizes of all living things remains unclear. We compile the present-day global body size-biomass spectra for the terrestrial, marine, and subterranean realms. To achieve this compilation, we pair existing and updated biomass estimates with previously uncatalogued body size ranges across all free-living biological groups. These data show that many biological groups share similar ranges of body sizes, and no single group dominates size ranges where cumulative biomass is highest. We then propagate biomass and size uncertainties and provide statistical descriptions of body size-biomass spectra across and within major habitat realms. Power laws show exponentially decreasing abundance (exponent -0.9±0.02 S.D., $R^2 = 0.97$) and nearly equal biomass (exponent 0.09 ±0.01, $R^2 = 0.56$) across log size bins, which resemble previous aquatic size spectra results but with greater organismal inclusivity and global coverage. In contrast, a bimodal Gaussian mixture model describes the biomass pattern better ($R^2 = 0.86$) and suggests small (~$10^{-15}$ g) and large (~$10^7$ g) organisms outweigh other sizes by one order magnitude (15 and 65 Gt versus ~1 Gt per log size). The results suggest that the global body size-biomass relationships is bimodal, but substantial one-to-two orders-of-magnitude uncertainty mean that additional data will be needed to clarify whether global-scale universal constraints or local forces shape these patterns.

## Introduction

Body size is a widely used metric in biodiversity, ecological, and evolutionary sciences because it is understood to mechanistically link physical, physiological and demographic processes [1,2]. Organisms on Earth range from $10^{-17}$ (*Nanoarchaeum equitans*) to $10^9$ g (*Sequoiadendron giganteum*) in body size when estimated as carbon weight. Body size representations within various taxa have been a major focus in macroecology and biogeography. Such representations are called size spectra, with size-biomass spectra being the cumulative biomass of selected organisms distributed across body size classes, integrated over all individuals and taxa (i.e., not averaging over species). These spectra are also known as biomass size spectra, which are related to size-abundance or normalized size-biomass spectra [3] (see Table 1 for summary of key terms).

Alliance (ET & MP); and NSF awards OCE-1426891, DEB-1616821, and OISE-1743711 (MP). The funders had no role in study design, data collection and analysis, decision to publish, or preparation of the manuscript.

**Competing interests:** The authors have declared that no competing interests exist.

**Table 1. Key terms.**

| Statistic | Definition |
|---|---|
| Normalized biomass ($B_N$) | $B$ normalized by the width of the body size class. For example, with width defined as one order of magnitude, $B_N = B/(10^{x+0.5}-10^{x-0.5})$ and $\log_{10}B_N = \log_{10}B - x - 0.454$ |
| Size-biomass spectra | $\log_{10}(B)$ per unit $x$ |
| Normalized size-biomass spectra | $\log_{10}(B_N)$ per unit $x$ |
| Size-abundance spectra | $\log_{10}(B/x)$ per unit $x$ |
| Power law | 2-parameter linear model on log-log scale (exponent is slope $\alpha$ or $\beta$) |
| Gaussian mixture distribution | $n$ x 3-parameter model with $n$ superimposed Gaussian distributions |
| Generalized extreme value distribution | 3-parameter model for distributions with left or right-skew |
| Uniform distribution | 2-parameter models specifying the same probability across a range from minimum to maximum |
| Truncated distribution | A distribution that specifies zero probability outside of minimum and maximum sizes |

B is biomass [g], BN is normalized biomass [unitless], and x is log10 body size [g/g].

Theories have attempted to predict and explain size-biomass spectra in terms of energy availability and transfer, species interactions, metabolic scaling, and aquatic trophic structure [4–9]. Such theories have been applied within limited taxonomic ranges, especially for the relationships between body size and abundance in terrestrial mammalian herbivores [10], marine phytoplankton [11], cross-realm producers [12], and marine trophic communities [4,9]. Within groups that share an energy source (not necessarily with trophic links), energetic equivalence (equal energetic availability to all populations) predicts a power law exponent of -0.75 for size-abundance or size-normalized biomass spectra (where biomass is divided by the size class or bin width), or an exponent of 0.25 for size-biomass spectra [10,13] (Table 1). However, empirical studies show that substantial residuals exist within groups and that the exponent deviates across groups [14]. Across trophic levels, size-ordered predator-prey interactions (especially in aquatic communities) can lead to a power law exponent of -1 or less for size-abundance or normalized size-biomass spectra, which is equivalent to an exponent of 0 or less for size-biomass spectra [8,15–17]. Beyond fundamental science, the power law exponents have also been considered as indices of productivity among marine ecosystems [18]. Deviations from expected exponents can be used to understand perturbations to ecosystems, such as inferring changing food web structure and fish biomass due to fishing [3,19–21], or inferring changes to the real breadth of the energetic base in coral reef systems [22]. Thus, size spectra are important for understanding biological and anthropogenic constraints to life within biological communities.

Despite this progress on power laws, important questions remain about whether small, medium, or large organisms dominate standing biomass of life on Earth at the global scale [8,21,23]. Different disciplines have proposed different biomass modes with or without reference to power laws. From a microbiology or marine perspective, microbes appear to dominate life [9,24]. From the terrestrial perspective, large plants dominate [25]. Each has a legitimate claim based on analysis of particular ecosystems or sets of taxa, but these approaches also prevent a different and novel synthesis in which traditionally excluded organisms may fit in. Empirical studies of size-biomass relationships have yet to include both terrestrial producers and consumers, or both small and large marine producers. The common phrase of bacteria-to-whale, meant to convey a complete marine size range [2,3,9], actually leaves out macroalgae, seagrass, hard corals, and mangroves that have maximum sizes near that of blue whales.

Increased inclusivity could reveal deviations from previous theoretical assumptions about size-structured trophic communities that lead to power law predictions. However, macroecological power laws themselves first arose from empirical relationships [4,10,26,27], which only later inspired still-evolving theoretical explanations [7,28]. The fact that some organisms, habitats, and parts of biological materials are routinely excluded from macroecology suggests these entities are poorly understood and a larger picture is missing. Revealing global patterns is a key step towards understanding universal constraints. For example, metabolic and biochemical theories predict universal constraints that govern how biological rates vary with body size and temperature across all organisms, which are largely independent of between-organism interactions and habitat variations [28,29]. Inspiring and testing theories on biomass distributions at biome scales will depend on assessing the current state of living things, but this empirical exercise has so far been prevented by a lack of data synthesis on body size itself.

Our objective here is to compile the first global and taxonomically inclusive size-biomass spectra of present-day terrestrial, marine, and subterranean realms. Specifically, we compile—for the first time—data on body size range within major biological groups that include all free-living organisms. The groups we use are not strictly taxonomically consistent, but they are functionally meaningful and follow the convention of our main biomass data source [30]. We then offer statistical descriptions (Table 1) of the global and habitat realm-specific spectra and their uncertainties. Our statistical tests focused on pattern detection rather than on previous theoretical hypotheses because these do not directly apply to global size-biomass spectra. Both the methodology of size spectra construction and statistical analyses serve as guides for how to integrate a taxonomically inclusive set of data with substantial uncertainties. The resulting catalogue of biomass data matched to body sizes stands as a record of present knowledge about life on Earth. We then focus on assessing the quality of available data in order to guide future research on causal mechanisms.

## Results

The body sizes (Tables 2–4) that comprise the most biomass on Earth are the small (mainly bacteria and archaea, $10^{-15}$ g per individual) and the large (mainly plants, $10^7$ g), and these peaks (15 Gt and 65 Gt per log size) outweighed intermediate sizes ($10^{-11}$ g to $10^{-2}$ g, ~1 Gt) by an order of magnitude (Fig 1A). The pattern is particularly clear on a linear biomass scale (Fig 1B). Biomass uncertainty persisted across all sizes, with 95% confidence bounds being two orders of magnitude from the smallest size to about 10 g and about one order of magnitude at larger sizes. Multiple unrelated groups exhibited similar upper size limits, including forest plants, grassland plants, fungi, wild terrestrial mammals, mangroves, fish, hard corals, seagrass, and marine mammals that contribute to the cumulative biomass peak at the size of $10^7$ g. All data and code are provided at https://github.com/EWTekwa/BodySizeBiomass.

Our inferred within-group size-biomass relationships (Fig 2) appear reasonable, with fish and plant spectra being comparable to previous community-level results that are relatively well-studied [8,100]. Total biomass in the smallest size classes ($<10^{-16}$ g) is dominated by marine bacteria (Fig 2AA). The biomass peak around $10^{-15}$ g is dominated by subterranean bacteria (Fig 2AH). Next, terrestrial fungi top the size range of $10^{-12}$ g to 1 g (Fig 2AG). Finally, grassland plants (1 g to 10 g, Fig 2GI) and forest plants (10 g to $10^9$ g, Fig 2GJ) make up almost all remaining biomass. We note that mangroves, hard corals, macroalgae, and seagrass make up 45% of total marine biomass even though they have been ignored in previous size spectra studies [2,3,9].

Terrestrial and marine spectra are different. Large body sizes dominate on land and across habitat realms, while the marine spectrum is roughly even across sizes (Fig 3). Marine

**Table 2. Terrestrial body sizes and biomasses.**

| Group | Smallest | Largest | Min. body size (g C) | Median body size (g C) | Max. body size (g C) | Biomass (Gt C) | Uncertainty (fold) |
|---|---|---|---|---|---|---|---|
| *Producers* | | | | | | | |
| Forest plants | *Salix herbacea*[*] | *Sequoiadendron giganteum* | 10.8 [31,32] | $1.13 \times 10^6$ | $2.24 \times 10^9$ [33] | 337.5 [31] | 1.2 |
| Grassland plants | *Mibora minima* | *Holcus mollis* | $3.75 \times 10^{-3}$ [34] | $4.32 \times 10^6$ | $1.34 \times 10^9$ [31] | 112.5 [31] | 1.2 |
| Cryptogamic phototrophs | *Nostoc punctiforme* | *Dawsonia superba*[†] | $1.15 \times 10^{-11}$ [35] | $2.72 \times 10^{-10}$ [a] | 87.5 [36] | 2.5 [b] | 2 |
| *Consumers* | | | | | | | |
| Soil bacteria | *Actinobacteria spp.* | *Proteobacteria spp.* [*] | $7.37 \times 10^{-16}$ [37] | $2.86 \times 10^{-14}$ | $1.15 \times 10^{-11}$ [37] | 7.352 | 6 |
| Soil archaea | *Crenarchaeota spp.*[*] | *Crenarchaeota spp.* [*] | $7.37 \times 10^{-16}$ [37] | $2.91 \times 10^{-14}$ | $4.72 \times 10^{-14}$ [37] | 0.516 | 4 |
| Soil protists | *Myamoeba spp.* [o] | *Dictyamoeba spp.*[*] | $7.37 \times 10^{-13}$ [38] | $7.37 \times 10^{-13}$ | $5.03 \times 10^{-11}$ [39] | 1.605 | 4 |
| Soil fungi | *Batrachochytrium dendrohabditis*[*] | *Armillaria ostoyae* | $7.37 \times 10^{-13}$ [40] | $1.53 \times 10^{-11}$ | $9.70 \times 10^6$ [41] | 11.802 | 3 |
| Terrestrial arthropods | *Archegozetes longisetosus* | *Birgus latro* | $1.50 \times 10^{-5}$ [42] | $2.00 \times 10^{-4}$ | $6.00 \times 10^2$ [43] | 0.212 | 15 |
| Humans | *Homo sapiens* | *Homo sapiens* | $3.75 \times 10^3$ [44] | $8.13 \times 10^3$ | $1.13 \times 10^4$ [44] | 0.055 | 1.1 |
| Livestock | *Gallus gallus domesticus* | *Bos taurus* | 270 [30] | $2.08 \times 10^4$ | $2.25 \times 10^5$ [30] | 0.107 | 1.1 |
| Wild land mammals | *Craseonycteris thonglongyai* | *Loxodonta africana* | 0.038 [45] | $2.53 \times 10^3$ | $1.65 \times 106$ [46] | 0.003 | 4 |
| Terrestrial nematodes | *Protohabditis hortulana*[†] | *Unspecified species*[†] | $6.02 \times 10^{-13}$ [47] | $5.00 \times 10^{-8}$ | $7.74 \times 10^{-8}$ [48] | 0.002 | 10 |
| Wild birds | *Mellisuga helenae* | *Struthio camelus* | 0.27 [49] | 6.67 | $1.50 \times 10^4$ [50] | 0.199 | 10 |
| Annelids | *Dendrobaena mammalis*[†] | *Microchaetus rappi* | $4.16 \times 10^{-8}$ [14] | $2.59 \times 10^{-4}$ | $2.25 \times 10^2$ [51] | 0.006 | 10 |
| Reptiles | *Brookseia spp.* | *Crocodylus porosus* | 0.027 [52] | $1.05 \times 10^2$ | $1.80 \times 10^5$ [53] | 0.003 | 100 |
| Amphibians | *Paedophryne amauensis* | *Andrias davidianus* | 0.003 [54] | 1.00 | $7.50 \times 10^3$ [55] | 0.001 [c] | 100 |

[*] indicates spherical bodies formula ([56] for microbes), and [†] indicates tubular bodies formula ([57] for microbes). Biomass and uncertainty are from [30] unless indicated. Alphabetical superscripts refer to Notes on Biomass and Body Size Calculation in Methods and Materials.

organisms may only contribute significantly to the global biomass spectrum at the size range of $10^{-12}$ g to $10^{-3}$ g and below $10^{-16}$ g. Marine biomass is overall likely dwarfed by terrestrial and subterranean biomass, though there is higher uncertainty in total biomass across size classes in the marine realm when compared to the terrestrial realm.

Linear regression of log biomass on log body size indicates a global power exponent $\beta$ of 0.086±0.001 (s.d. across bootstraps) with a mean $R^2$ of 0.56 (Fig 4A). For the terrestrial realm, we obtained a similar $\beta$ of 0.100±0.008 with a mean $R^2$ of 0.66 (Fig 4F). These results show that biomass increases with size. Even though the variances explained are high, these power laws fail at the small size range, with confidence bounds missing the size class with the most biomass, filled by microbes. For the marine realm we obtained a much lower $\beta$ of 0.019±0.005 with a mean $R^2$ of 0.11, indicating a similar biomass across log size bins (Fig 4K).

The overall and terrestrial spectra show similar small mean power law exponents $\beta$ (0.051 to 0.086 and 0.047 to 0.100 respectively), while the marine spectrum has an effectively zero $\beta$ (-0.007 to 0.022) across choices of within group truncation methods, use of ramets (physiological individuals) instead of genets (colonies of genetically identical individuals) as body sizes, and exclusion of metabolically inactive biomass like subterranean microbes (Table 5, S1 Fig). If the linear regressions were performed on log size-log abundance instead (equivalent to normalized size-biomass spectra), we would obtain exponents $\alpha$ of -0.90±0.02 ($R^2 = 0.98$), -0.80 ±0.05 ($R^2 = 0.88$), and -0.96±0.03 ($R^2 = 0.98$), which are approximately $\beta$-1 as abundance is biomass divided by size (but not exactly because the data, not the mean exponents, were directly transformed, S2 Fig). As the inflated $R^2$ suggest, the transformation from biomass to

**Table 3. Marine body sizes.**

| Group | Smallest | Largest | Min. body size (g C) | Median body size (g C) | Max. body size (g C) | Biomass (Gt C) | Uncertainty (fold) |
|---|---|---|---|---|---|---|---|
| *Producers* | | | | | | | |
| Mangroves | *Rhizophora mangle** *(dwarf)* | *Rhizophora mangle** *(canopy)* | $4.06 \times 10^4$ [58] | $6.49 \times 10^5$ [d] | $2.88 \times 10^7$ [58] | 3.5 [59] | 1.4 |
| Seagrass | *Halophila decipiens** | *Posidonia oceanica** | $2.63 \times 10^{-3}$ [60] | $7.53 \times 10^4$ [e] | $6.91 \times 10^7$ [61,62] | 0.11 | 10 |
| Macroalgae | *Phaeophyceae spp.* | *Macrocystis pyrifera* | 0.135 [63,64] | 2.00 [f] | $2.70 \times 10^3$ [63,64] | 0.14 | 10 |
| Bacterial picophytoplankton | *Prochlorococcus spp.* | - | $5.00 \times 10^{-14}$ [65,66] | $9.13 \times 10^{-14}$ [g] | $1.67 \times 10^{-13}$ [h] | 0.13 | 10 |
| Green algae / protist picophyto-plankton | *Ostreococcus tauri* | - | $1.05 \times 10^{-13}$ [65,67] | $1.49 \times 10^{-13}$ [i] | $2.10 \times 10^{-13}$ [j] | 0.30 | 10 |
| Diatoms | *Thalassiosira pseudonana* | *Ethmodiscus spp.* | $2.4 \times 10^{-11}$ [68] | $9.08 \times 10^{-9}$ [k] | $5.11 \times 10^{-6}$ [68] | 0.31 | 10 |
| Phaeocystis | *Phaeocystis globosa cell* [o] | *Phaeocystis globosa colony** | $1.15 \times 10^{-11}$ [69] | $5.24 \times 10^{-4}$ [l] | 0.047 [69] | 0.28 | 10 |
| *Consumers* | | | | | | | |
| Marine bacteria | *Pelagibacter ubique** | *Thiomargarita namibiensis** | $5.50 \times 10^{-16}$ [70] | $1.32 \times 10^{-14}$ | $1.10 \times 10^{-4}$ [71] | 1.327 | 1.8 |
| Marine archaea | *Nanoarchaeum equitans* | *Staphylothermus marinus** | $1.47 \times 10^{-17}$ [72] | $1.22 \times 10^{-14}$ | $9.90 \times 10^{-11}$ [73] | 0.332 | 3 |
| Marine protists | *Picomonas judraskeda** | *Rhizarian spp.** | $1.44 \times 10^{-12}$ [74] | $2.26 \times 10^{-12}$ | $7.37 \times 10^{-4}$ [75] | 1.058 | 10 |
| Marine arthropods | *Stygotantulus Stocki* | *Homarus americanus* | $3.537 \times 10^{-8}$ [42,43] | $7.08 \times 10^{-6}$ | $3.00 \times 10^3$ [76] | 0.940 | 10 |
| Fish | *Paedocypris progenetica* | *Rhincodon typus* | $1.50 \times 10^{-4}$ [77] | 0.627 | $4.63 \times 10^6$ [78] | 0.668 | 8 |
| Molluscs | *Ammonicera minortalis* | *Mesonychoteuthis hamiltoni* | 0.01 [79,80] | $4.02 \times 10^{-4}$ | $3.98 \times 10^4$ [81–83] | 0.182 | 10 |
| Cnidaria | *Psammohydra nanna* | *Cyanea capillata* | $1.00 \times 10^{-5}$ [84,85] | $5.09 \times 10^{-3}$ | $1.00 \times 10^5$ [84,86] | 0.040 | 10 |
| Hard corals | *Leptopsammia pruvoti* [m] | *Porites lutea* | 6.41 [87,88] | $1.54 \times 10^3$ [n] | $1.68 \times 10^7$ [89] | 0.653 [°] | 4 |
| Wild marine mammals | *Arctocephalus townsendi* | *Balaenoptera musculus* | $4.05 \times 10^3$ [90] | $7.42 \times 10^4$ | $2.99 \times 10^7$ [78] | 0.004 | 1.4 |
| Marine nematodes | *Thalassomonhystera spp.* | *Platycomopsis spp.* | $7.50 \times 10^{-9}$ [91] | $1.80 \times 10^{-7}$ [91] | $1.20 \times 10^{-5}$ [91] | 0.014 | 10 |
| Marine fungi | *Malassezia restricta* | *Penicillium chrysogenum* | $5.89 \times 10^{-12}$ [92,93] | $1.39 \times 10^{-11}$ | $1.89 \times 10^{-5}$ [94] | 0.325 | 10 |

* indicates spherical bodies formula ([56] for microbes). Biomass and uncertainty are from [30] unless indicated. Alphabetical superscripts refer to Notes on Biomass and Body Size Calculation in Methods and Materials.

abundance may lead us to conclude that there is roughly equal biomass across all sizes (or slightly higher at large sizes on land), and there are little deviations visible from the power laws (S2 Fig). In comparison, the size-biomass spectra (Fig 4) are roughly detrended versions of

**Table 4. Subterranean consumer body sizes.**

| Group | Smallest body size | Largest body size | Min. body size (g C) | Median body size (g C) | Max. body size (g C) | Biomass (Gt C) | Uncertainty (fold) |
|---|---|---|---|---|---|---|---|
| Subterranean bacteria | *Proteobacteria spp.* | *Desulforudis audaxviator* | $9.81 \times 10^{-16}$ [95] | $2.1 \times 10^{-14}$ [96] | $5.90 \times 10^{-12}$ [97] | 18.9 [p] | 3 [q] |
| Subterranean archaea | *Thermoproteus spp.* | *Miscellaneous Crenarchaeotal Group spp.* | $2.49 \times 10^{-15}$ [98] | $2.1 \times 10^{-14}$ [96] | $9.22 \times 10^{-14}$ [99] | 8.1 [r] | 3 [s] |

Alphabetical superscripts refer to Notes on Biomass and Body Size Calculation in Methods and Materials.

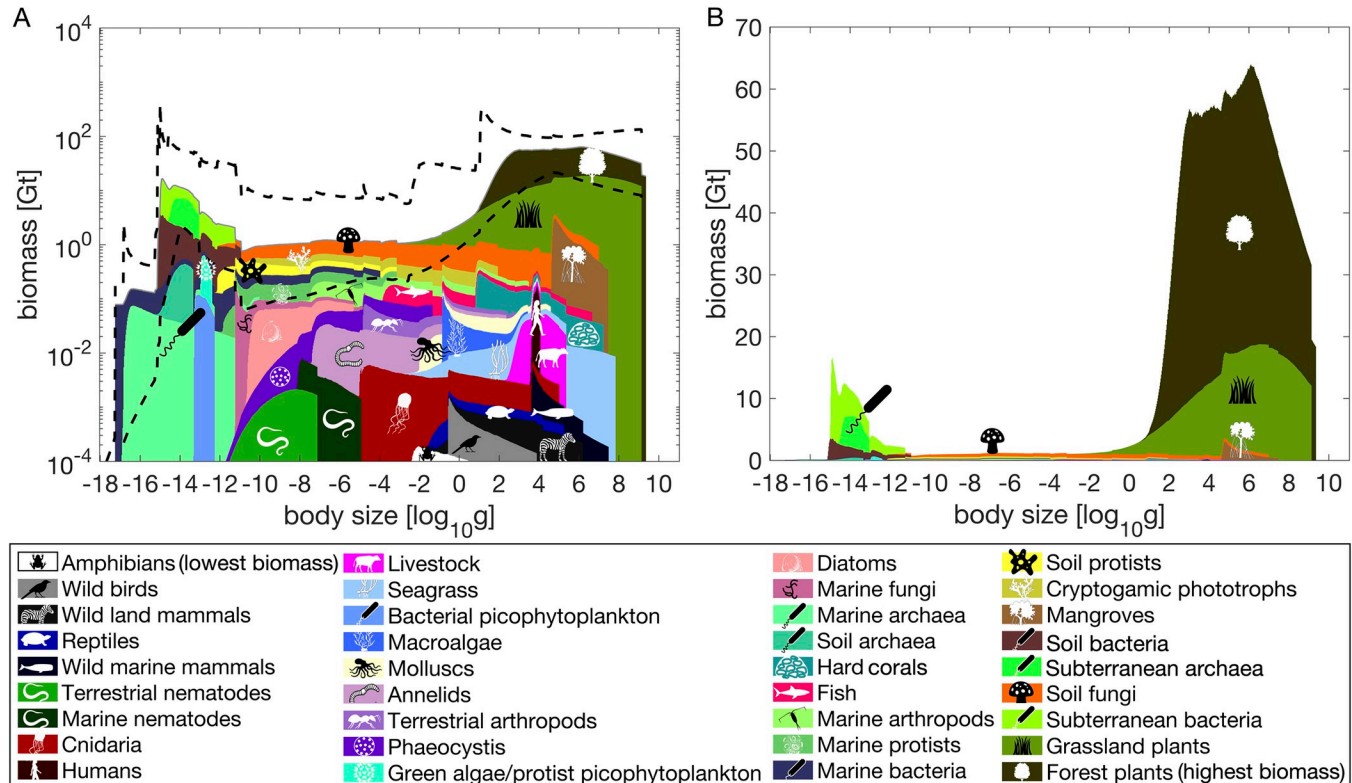

**Fig 1. Global body size biomass spectrum.** A. Median carbon biomass (log scale) per log size as a function of body size with 95% confidence bounds (black dotted curves) cumulated across biological groups from 1000 bootstraps over within-group biomass and body size error distributions. Groups were organized from the least massive at the bottom to the most massive at the top for visibility on the log scale (ordered from top left to bottom right in color legend for group identity). Group biomasses are stacked so each group's biomass is represented by its upper y-axis location minus its lower y-axis location (not by the upper y-axis location alone). See Tables 2–4 for within-group biomass uncertainties, and S3 Table; for icon sources. B. Median biomass in linear biomass scale. Confidence bounds are not shown here because they are so large as to obscure the median patterns on the linear scale.

size-abundance, with the -1 slope between size and abundance being the "trivial" trend on top of which both linear (power laws) and nonlinear (multimodal) patterns emerge.

Across terrestrial, marine, and subterranean (under both land and sea) organisms, there is a consistent $\log_{10}$ ratio of maximum to minimum size (size range) across all groups regardless of median size (slope = 0, p = 0.99), with a mean ratio of 7.0±4.2 (S.D.). In other words, as mean size increases, size range also increases with a power law exponent of 0 (S3 Fig). This supports the view that the non-normalized size-biomass spectra are an appropriate way to investigate representation across size, in addition to the statistical reasons outlined above.

Gaussian mixture models capable of multiple biomass modes reveal decreasing AICc scores with increasing number of Gaussian components overall and within realms, indicating better statistical descriptions than power laws (linear regressions) (Fig 4B–4E, 4G–4J and 4L–4O). However, visual inspection suggests the size-biomass relationships are well described by two mixture components, and further complexities appear hard to substantiate given the spectral uncertainty and variations in AICc across bootstraps (Fig 4C, 4H and 4M). These two-mode regressions explain much more of the data variation ($R^2$ = 0.86, 0.84, and 0.56 for all realms, terrestrial, and marine respectively) than power laws, the main difference being the ability to identify both small and large size-biomass modes. These results indicate two size modes are important and useful description of the global biomass spectra, beyond simple power laws.

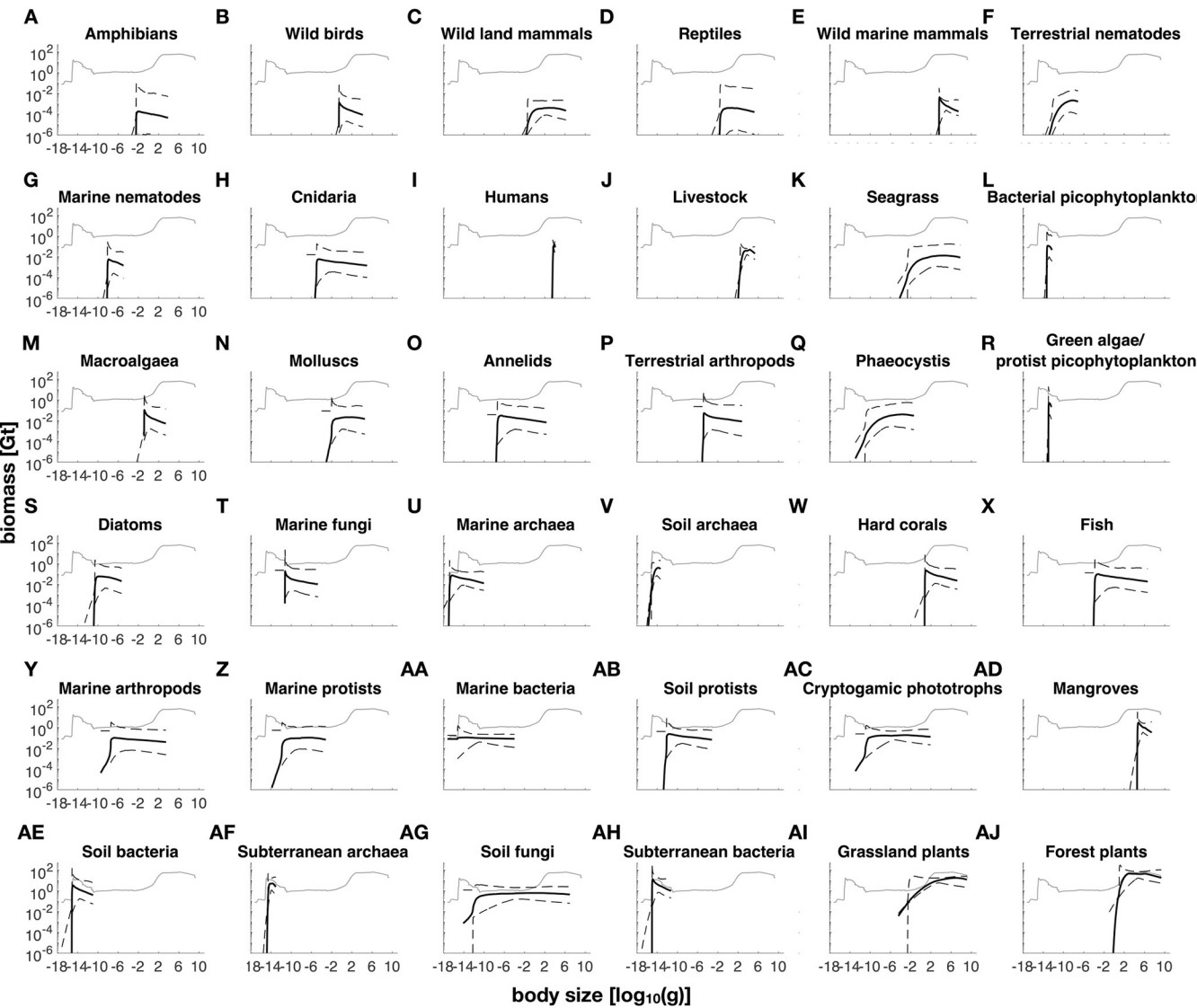

**Fig 2. Body size biomass spectra within groups.** Thick black curve is the median log biomass, and black dotted curves are 95% confidence bounds from 1000 resamples from within-group size and biomass uncertainties. Groups are organized from lowest to highest biomass (A to AJ). For reference, the thin grey curve is the median cumulative log biomass of all groups.

## Discussion

We performed a novel synthesis of the mass of all life in the biosphere, revealing size-biomass patterns that contain features reminiscent of published results [4,8,9,20,21], but also new features attributable to a greater taxonomic and error inclusion than previous efforts. Our three major biological findings were: 1.) lower and upper size limits were shared by diverse organisms, and these extreme sizes appear to contain most of the biomass on Earth; 2.) there was relatively consistent biomass across log body size classes, described by power law exponents near zero; and 3.) there was a greater proportion of total biomass on land concentrated in large organisms when compared to the ocean. Methodologically, we found that analyses relating log-biomass to log-size bins across all organisms (rather than size-abundance or normalized size-abundance), while retaining uncertainties in both size and biomass, revealed the most nuanced patterns.

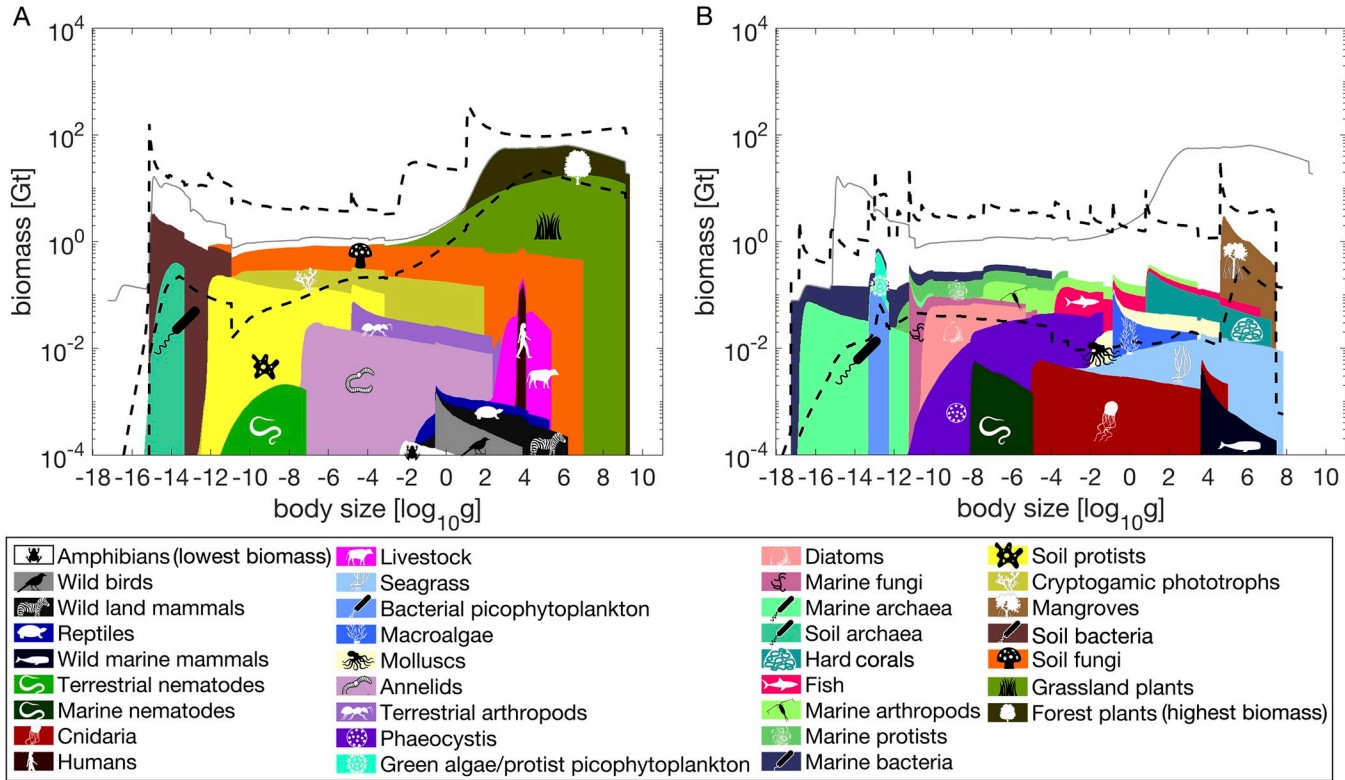

**Fig 3. Body size biomass spectra by habitat realms.** See Fig 1 caption for description. A. Terrestrial. B. Marine. Subterranean prokaryotes are excluded. Thin grey curves are the median cumulative log biomass of the global biome.

The first pattern indicates near-universal lower and upper size limits where the highest biomass accumulates. It is well-known that bacteria and archaea would share the lower size limit of all living things at around $10^{-17}$ to $10^{-16}$ g. More surprisingly, multiple producer and consumer groups on land and in the sea coincide with maximum body sizes between $10^7$ and $10^9$ g–a relatively narrow range compared to the 26 orders of magnitude spanning all free-living things–including such diverse organisms as *Sequoiadendron giganteum*, *Holcus mollis*, *Armillaria ostoyae*, *Rhizophora mangleo*, *Posidonia oceanicao*, *Porites lutea*, and *Balaenoptera musculus*. This coincidence suggests an underlying upper size constraint, but multiple mechanisms may simply coincide [101,102]. Gaussian mixtures with two components describe size-biomass spectra better than power laws across-realm and within terrestrial and marine realms, again showing that the lower and upper size limits across all free-living things are also modes where biomass is most concentrated. While our mean estimates indicate these modes contain roughly one order magnitude more biomass per log size than intermediate body sizes, uncertainty in biomass was consistently higher than this magnitude, indicating that the data is too poorly resolved to unequivocally support the bimodal pattern.

The second pattern indicates similar biomass across a large size range (a zero power law exponent explaining how biomass varies with body size). This is highly consistent with size spectra documented for aquatic ecosystems or within some taxonomic groups [4,10,13,28], which supports metabolic, competitive, and trophic explanations [17,28]. However, unlike previous studies, we included microbes, large producers, and other traditionally excluded marine groups summing to 45% of total marine biomass [2,3,9]., and propagated both biomass and size uncertainties. The fact that a near-zero exponent still persisted across all habitat realms and analytical assumptions is surprising because our global-scale patterns are not likely

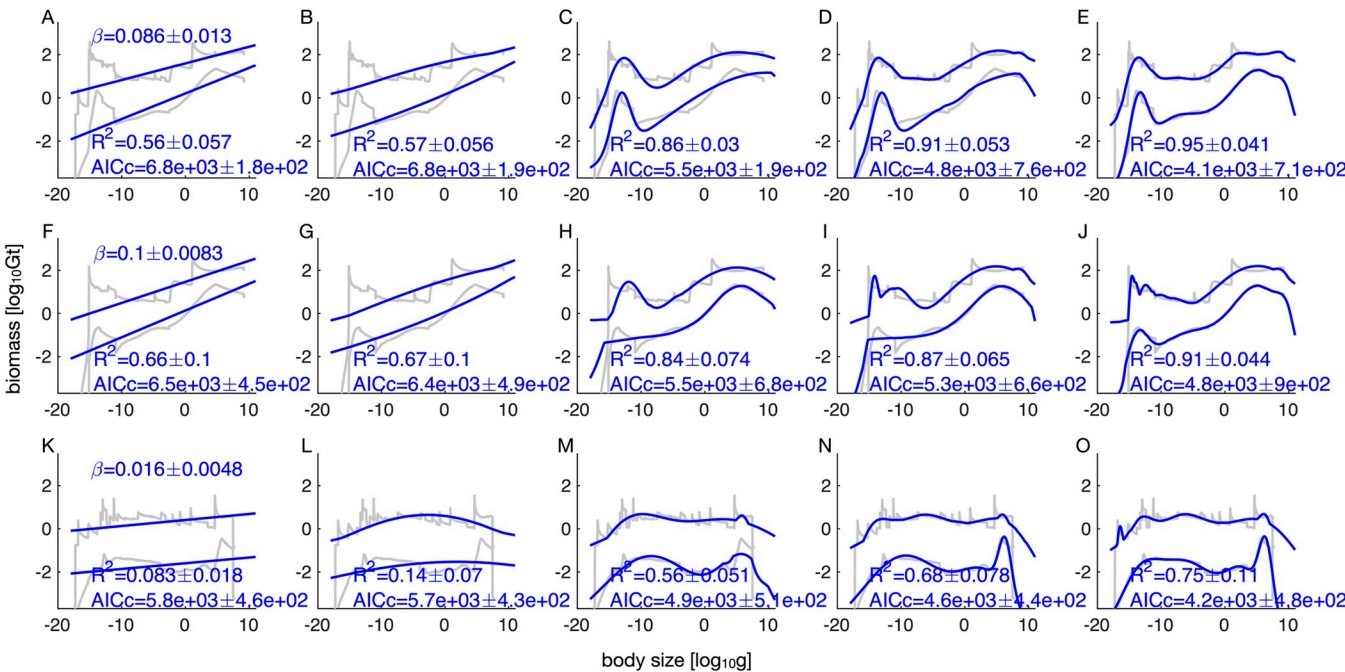

**Fig 4. Regression analyses.** Rows represent habitat realms (A to E: All realms, F to J: Terrestrial, K to O: Marine). Columns represent regression model types: (A, F, K: Linear, B, G, L: Gaussian, C, H, M: Gaussian mixture 2, D, I, N: Gaussian mixture 3, E, J, O: Gaussian mixture 4). Grey curves represent 95% confidence intervals of the data, and blue curves represent 95% confidence intervals of the model from 1000 bootstraps. For linear models, regression slopes are mean power exponents ± standard deviations across bootstraps. $R^2$ and AICc scores are means ± standard deviations across 1000 bootstraps.

shaped by interactive forces such as trophic or competitive interactions previously proposed to cause near-zero exponents [17]. We found some evidence for bimodality that diverged from power laws, but large uncertainties prevent clear conclusions about whether or why such non-linear patterns occur.

The third finding, that biomass in the ocean is somewhat more evenly distributed across size classes than on land offers clues to a future theoretical synthesis. The marine realm exhibits trophic positions roughly determined by body size, thus the marine spectrum conforms closer to a trophic-mediated uniform log-log size-biomass expectation [21,28]. Biophysics and ecology—competition for nutrients—explain why primary producers are small in the ocean versus large on land [4,103,104]. However, this narrative overlooks the striking similarities between the two realms. Large primary producers that also provide physical structures to ecosystems dominate both land and sea (grass, tree, mangroves, corals, seagrass and kelps). Despite their large biomass, however, we note that large marine primary producers are restricted to shallow seas in which access to light and nutrients in the sediment create a biophysical environment part way between ocean and land, do not dominate all marine ecosystems (e.g., pelagic), and may be considered its own realm. In addition, excluding "metabolically inactive" material such as wood, subterranean microbes, and skeleton produced by living corals would flatten the size-biomass spectra globally and in both terrestrial and marine realms (closer to $\beta = 0$, Table 5), but without erasing the apparent global bimodality and differences across realms (S1D Fig). The causes of size-biomass differences in different habitat realms remain to be explored.

Together, the findings of universal size limits possibly coinciding with a bimodal biomass distribution, overall similar biomass across sizes, and differences between habitat realms suggest possible roles for both universal and local explanations, depending on which feature of

**Table 5. Size-biomass power law exponents across realms and assumptions.** Assumptions correspond to sensitive analyses plotted in S1 Fig. Exponents and $R^2$ result from 1000 bootstrapped linear regressions of log biomass on log size.

| | $\beta$ exponent (± bootstrap S.D.) | | | $R^2$ (± bootstrap S.D.) | | |
| --- | --- | --- | --- | --- | --- | --- |
| *Realm*<br>Assumptions | *All* | *Terrestrial* | *Marine* | *All* | *Terrestrial* | *Marine* |
| **A.** All free-living, body size cutoff at -2/+0 $\log_{10}$g of reported (base model) | 0.086±0.013 | 0.100±0.008 | 0.016±0.005 | 0.56±0.06 | 0.66±0.10 | 0.08±0.02 |
| **B.** All free-living, body size cutoff at ±1 $\log_{10}$g of reported | 0.082±0.007 | 0.079±0.007 | 0.019±0.005 | 0.40±0.08 | 0.45±0.11 | 0.05±0.03 |
| **C.** All free-living, body size cutoff at ±0 $\log_{10}$g of reported | 0.082±0.013 | 0.087±0.017 | 0.020±0.002 | 0.55±0.06 | 0.70±0.07 | 0.13±0.04 |
| **D.** Ramet size definition, body size cutoff at -2/+0 $\log_{10}$g of reported | 0.083±0.012 | 0.097±0.008 | 0.016±0.005 | 0.58±0.07 | 0.66±0.11 | 0.09±0.02 |
| **E.** Metabolically active mass only, body size cutoff at -2/+0 $\log_{10}$g of reported | 0.078±0.016 | 0.079±0.010 | -0.009±0.006 | 0.68±0.09 | 0.58±0.13 | 0.05±0.03 |

size-biomass spectra we focus on. Previously unexplored universal constraints, perhaps similar to known biochemical [29] or spatial-cellular mechanisms [105], can conceivably explain size limits and multiple high-biomass modes at different sizes, but these constraints may be modified or overwritten by local interactions between different organisms at finer spatial scales. The relative strengths of universal versus local constraints may be partially understood by comparing size-biomass spectra and their uncertainties across-realm versus within-realm. For instance, if the multiple modes observed across-realm are shared by different realms, then spectral uncertainties should be lower across-realm because of more data (lower observation error and greater taxonomic coverage [106,107]) and universal constraints may be responsible. On the other hand, if different realms contribute different size modes, then spectral uncertainties should be higher for the across-realm spectrum because of higher biological variance, supporting the hypothesis that local constraints likely shape the across-realm pattern. However, this reduction in uncertainties at smaller scales is only detectable if sample coverage does not drastically decrease. In our analyses, some size modes coincide across all realms, leaving for the possibilities of both universal and local constraints. In addition, the across-realm data exhibits narrower confidence bounds and a stronger signal of bimodality than the terrestrial realm alone (Fig 4H and 4M), and even more so when compared to the relatively hard-to-sample marine realm alone, because of higher aggregate data availability. These mode overlaps and uncertainty patterns indicate that universal constraints may strongly shape size-biomass spectra everywhere in similar ways, but this impression may also be due to a lack of data.

Our study shows that body size biomass spectra include substantial uncertainties. Within-group biomass uncertainties are high among some taxa, especially in microbes [30]. Data and synthesis of within and between-study uncertainties on biomass that we base our study on remain crude across groups [30] but are consistent with estimates from independent studies on plant and fish [8,100,108]. We have also filled the important gaps of marine habitat builders [59,88,109,110] and incorporated latest estimates for subterranean microbes [96,111]. Definitions of body size (ramets vs. genets), mass (with vs. without metabolically inactive components like wood, skeleton, and subterranean microbes), and realm (mangroves being marine, terrestrial, or partial) remain open for debate. Sensitivity analyses of these variations on cumulative size-biomass spectra show crude patterns like power laws are consistent, but nuances like the location of size-biomass peaks are uncertain. Our methodology was designed to minimize biases and propagated different sources of uncertainty. Indeed, this approach identified that large uncertainty persists through all sizes. In contrast, most previous macroecological studies have assumed certainty in minimum and maximum sizes (size classes) instead of propagating size error [4,7,9,12,112,113]. This assumption would have resulted in nearly uniform biomass distributions across log sizes within biological groups, which though did not affect

mean power law parameter estimates, severely underestimated biomass uncertainty particularly at large sizes. Intuition tells us we are nowhere near as certain about where biomass is concentrated at large sizes (1.2-fold uncertainty at sizes 10 to $10^9$ g assuming near-uniform within-group distributions in S1B Fig, which is just the total biomass uncertainty for plants independent of size). Error propagations in both size and biomass, as well as flexible within-group size-biomass distributions rather than strong assumptions like uniformity or a particular skew (like power law, Gaussian, or lognormal), result in ~10 fold uncertainty at the same size range (Fig 1). Given current knowledge on how size range varies with size within biological groups and how biomass varies across sizes, we recommend studying the relationship between log-biomass and log-size (i.e. size-biomass spectra) using both power laws and nonlinear statistics such as Gaussian mixtures. Our results highlight as much the current knowledge about the Earth's biosphere as it does potential gaps in observation. For instance, missing observations in specific size classes will tend to create an impression of multimodality even if in reality there is a continuum of biomass across sizes. Multiple within or between-study biomass estimates for particular biological groups may not be spatially independent and thus not representative, which can lead to an underestimation of uncertainty and bias in expected total biomass. However, we would not know what these uncertainties and biases are without more sampling. In light of these limitations, uncertainties of our knowledge of size-biomass spectra were likely underestimated (but to a less severe degree than other macroecologoical studies [4,7,9,12,28,112,113]), yet even these optimistic estimates reveal how little we know about our global biosphere. Quantifying uncertainties while identifying knowledge gaps remain priorities for macroecology [114].

The state and change of size-biomass spectra should be an urgent biodiversity assessment objective and a fertile ground for fundamental theories. The massive data requirement to conduct a more detailed spectral survey may resemble modern cosmology and its collaborative search for patterns in matter distribution [115]. Our results provide a first crude roadmap for what patterns may exist, but they will likely drastically change if size-biomass spectra become targets for research programs. Moving forward, macroecology should embrace taxonomic inclusivity and unexplored scales that defy existing explanations.

## Materials and methods

### Biomass data

To compile the global aggregate body size biomass spectrum among biological groups defined by habitat and taxonomy, we used global biomass (gigatons [Gt] in carbon content) assessments and minimum, median, and maximum body sizes (grams [g] in carbon content) within groups (Tables 2–4). We started with the most comprehensive existing synthesis of global biomass estimates, which incorporate uncertainties within and between multiple studies [30]. We followed the biological grouping in Bar-On's database [30], which is not at a consistent taxonomic level but instead reflect the highest resolution at which a biomass estimate is available and comparable to other groups. Bar-On et al. drew from hundreds of studies that reported either biomass per sampled area or global extrapolations. The biomass per sampled area data was extrapolated by Bar-On et al. to the global scale based on the spatial distribution of environmental variables such as temperature and habitat type (akin to species distribution models but at a higher taxonomic level). The best estimates were obtained from the geometric mean of multiple data sources within group, and within- and between-study uncertainties were propagated (S4 Fig; see Bar-On et al.'s supplementary). We recognize that estimates of mean biomass and uncertainty can likely be improved for all groups, but this is not the main goal of our paper. Instead, we complemented Bar-On's database only when biological groups with

potentially high biomass were missing or clearly outdated, including cryptogamic phototrophs [109], hard corals [88,110], mangroves [59], and subterranean prokaryotes [96,111]. We placed mangroves in the marine realm because they live in coastal salt water, support a high diversity of marine fish, and are considered an integral part of blue carbon accounting [58,59,116]. Details for these new estimates are described in the footnotes of Tables 2–4. For some biological groups, new and potentially relevant data has appeared after Bar-On's publication. However, these studies cataloged only biomass by species without assessing their contributions to overall group biomass (e.g., bird [117] and mammals [118]), did not directly address present-day biomass (e.g., fish [119]), or were nearly identical to Bar-On's original estimates (e.g., terrestrial plants [108]). We included the plant woody material and coral skeleton produced by a living individual as part of biomass in our primary analysis, as was done in a previous global biomass synthesis [30]. This approach is consistent with the idea that all biomass regardless of metabolic status contributes to ecosystem functioning, though we also explored removing this biomass for sensitivity analyses and for future investigations.

## Body size data

Size was defined as the carbon content (grams) of a unicellular or multicellular organism. Defining an organism is not entirely straightforward for clonal life forms like grasses, corals, and fungi. Here, we used genets as our primary definition but also explore the consequences of using ramets to measure body size. Genet is a colony of genetically identical ramets in a location from a single parent, whereas a ramet is a physiologically distinguishable individual. Genet is a widely accepted functional definition of a biological unit because genetically identical cell agglomerates function as coherent units and actively share resources, and often seem like separate organisms only because the connecting tissues are invisible to us above the substrate [61,120,121]. We collected minimum, median, and maximum genet sizes from a literature search (Tables 2–4). Three points for biomass distribution within each group is minimalistic but, given our current knowledge of most groups, there are few other reliable size data to serve as additional reference points across each biological group. In the literature, mean sizes are often reported without specifying the species while assuming a log-normal size-biomass distribution [30], so we can record these mean sizes as median size in our dataset without transformations. In cases where no mean or median sizes were reported in [30], we used sizes mentioned in the literature as qualitatively representative species (those mentioned as most "common" or "widespread"), which are likely closer to the median rather than the mean size, given no a priori knowledge of the distribution. We used sizes at maturity because this is likely where biomass is concentrated within species [21], and because data are not available for most taxa on the contribution of spore or juvenile stages. However, our choices of body size cutoffs in subsequent estimates of within-group size-biomass spectra can approximate the biomass share of these immature sizes.

We converted all size observations to an estimate of mass in terms of carbon. The body sizes of some species were reported in units of grams carbon, but for many species we needed to extrapolate from wet or dry mass. When size estimates in the literature were reported in wet mass, we first searched the literature for a species-specific wet weight to grams carbon conversion. When a species-specific conversion was not available, we used the conversion from the closest relative within the taxon (see online repository tables). When taxon-specific conversions were not available, we assumed 30% dry mass per wet mass unit, and 50% carbon per dry mass unit following previous conventions [30]. In some cases, body size was reported in units of length (particularly among annelids, nematodes, and fishes). For these taxa, we found existing length to weight conversions for the species or the closest relative within the taxon. If body

size was reported in diameter, as was the case for most unicellular species, we found the volume assuming that the organism was either spherical [56] or tubular [57], and then found existing biovolume to biomass conversions for the species or the closest relative within the taxon. For hard corals, since each corallite or colony is often tightly packed among other units, we estimated that volume as the cube of the reported diameter. While some of these assumptions may introduce size errors that we do not explicitly track in our uncertainty analyses, the different plausible conversion factors are within an order of magnitude. This error magnitude is much smaller than the size ranges estimated for each biological group based on the uncertainties that we did track (Fig 2).

We excluded from our body size (dry carbon mass) any non-free-living disease organisms, which are mainly found within trematode, nematode, virus, bacterial, and fungal groups. Disease organisms tend to represent extreme body sizes within their groups and may have been double counted as host biomass, which present a special challenge to estimating within-group size-biomass distributions that we do not address here. It is likely that the total biomass of disease organisms is low both within hosts (3% or less) and together as a group (similar to wild birds, the second lowest biomass among free-living groups) [24,122] and thus should not appreciably affect the cross-taxa spectrum, even though parasites and microbiome-associated organisms may have disproportionate effects on the biomass of other organisms.

To determine how biomass should be tallied by size class, we assessed how a group's body size (mass) range (as directly observed from data) is related to median body size. A group's size range represents an aspect of biological variation within which organisms can be considered similar. If groups with larger sizes vary in size by the same magnitude (rather than same order of magnitude) as groups with smaller sizes (e.g., group #1 contains 1-10g organisms, group #2 contains 1001g-1010g organisms), then tallying biomass by log size bins would group together increasingly different organisms at large median sizes. This is the rationale for normalized size-biomass, which divides the measured biomass of a size class by the class's presumably artificial size range [8]. Conversely, if groups' size range increases as a power function of median size, then larger size classes conceivably contain larger size variations that represent similar organisms. In this case normalization does not seem necessary on biological basis, and the size-biomass spectrum relating log biomass to log size, as often assumed [123], is natural. We performed a linear regression of the ratio of $\log_{10}$ maximum size to log10 minimum size (from known species) on $\log_{10}$ median size across biological groups. A slope (power exponent) of 0 would support the use of size-biomass spectra without normalization.

### Notes on biomass and body size calculation

For Tables 2–4, calculations and references for within-group body size and biomass different from [30] are documented below. The notes are labelled by alphabetical superscripts.

a. Among lichens, likely the most abundant among cryptogams, we estimate that 87% contain phycobionts (*Trebouxia* 8–21 μm) [124] and 13% contain cyanobionts (*Nostoc punctiforme* 5 μm) [35]. This composition was used to estimate the mean body size.

b. The total lichen biomass and uncertainty were obtained from [109]; to obtain cryptogamic phototrophs' biomass, the fungal portion of lichen was subtracted out. Twenty percent of fungi species occur in lichens [125], so 20% of the total fungal biomass was subtracted from the lichen biomass to get the cryptogamic phototrophs' biomass.

c. Assumes amphibian habitat area is mainly rainforest, $5.50 \times 10^{12}$ m$^2$ [31], and 0.1 individual per m$^2$ (lower than [30]'s likely overestimate). Uncertainty is unknown, so copied from reptiles which is the taxon with the highest uncertainty.

d. *Rhizophora mangle*, similar to estimates for other typical species [126]

e. Based on genet size of *Zostera marina*, a widespread species [127] and carbon density [62].

f. Based on *Laminaria saccharina*, a widespread species [64].

g. Diameter corresponds to definition of picophytoplanktons (2 μm), and corresponding carbon content is based on conversion formulae from the smallest species.

h. Maximum sizes are estimated to correspond to the same deviation from the median size as minimum sizes are (on log scale).

i. Same method as for bacterial picophytoplankton.

j. Same method as for bacterial picophytoplankton.

k. Based on *Dactyliosolen fragilissimus* [68].

l. Mean size of colonies of *P. globosa* (2 mm) and *P. pouchetii* (1.5 mm), which are globally distributed and associated with bloom formation [69].

m. Classified as "generalist coral" for size estimate [110].

n. Mean colony size was estimated as the geometric mean of corallite or maximum colony sizes. Only maximum colony sizes were found across species and may contain several genets, hence the geometric mean. For each estimate, measures for four coral types were converted first to cubic volumes using 3D morphologies, assuming branching morphotype for "competitive" and "weedy" corals, and massive morphotype for "generalist" and "stress-tolerant" corals [110]. Each volume estimates were then converted to mass using type-specific skeletal densities [128], C per CaCO3, and weighted by global coral cover contributions [88].

o. Mean skeleton biomass was the geometric mean of two biomass estimates based on global coral cover having heights corresponding to either corallites or maximum colony sizes. Mean tissue biomass was 0.05 Gt with a 10 fold uncertainty [30]. Overall mean biomass was the sum of mean skeleton and tissue biomass, and overall uncertainty was obtained from assuming that the overall min/max correspond to the sum of min/max skeleton and tissue estimates.

p. Total subterranean microbial biomass was assumed to be the geometric mean of 23 to 31 PgC (which is 27 PgC) from [96]. 70% of microbial abundance is expected to be bacteria [129].

q. Range of total subterranean microbial cell count from four models in [96] was 1.6 to 11.2 x $10^{29}$, with a geometric mean of 4.2 x $10^{29}$. This range corresponds to a three-fold uncertainty, which is similar to bacteria and archaea groups in other habitat realms.

r. 30% of microbial abundance is expected to be archaea [129]. See note for bacterial biomass.

s. Same as uncertainty for subterranean bacteria.

## Within-group size-biomass spectra

We used the truncated generalized extreme value (GEV) distribution to infer the body size-biomass distribution (with size on a log scale) within biological groups (see S4 Fig for examples). The probability distribution function for biomass $y(x)$ in gigatonnes was written in term of log size $x$, with $B$ being the total biomass of the group, and the three parameters $\mu$, $\sigma$, and $\xi$

specifying the location, scale, and shape, respectively:

$$y(x) = B\frac{1}{\sigma}t(z)^{\xi+1}\exp(-t(x)) \tag{1}$$

$$t(x) = \begin{cases} \left(1 + \xi\left(\frac{x-\mu}{\sigma}\right)\right)^{-1/\xi} & if \ \xi \neq 0 \\ exp\left(\frac{-(x-\mu)}{\sigma}\right) & if \ \xi = 0 \end{cases} \tag{2}$$

We chose the GEV distribution because it is flexible, encompassing previously proposed body size- biomass relationships outlined below. Cross-taxa size-biomass relationships are often described using power laws, with positive [10,13] or negative [8,15–17] exponents resulting in extremely left or right-skewed distributions (where the body size with the maximum biomass is at the end of the size range). For plant communities where community-level size-biomass relationships are better documented than other groups, the right-skewed Weibull distribution was used [100], which is a special case of the GEV. On the other hand, empirical studies on size-species frequency distributions, though not easily translatable to size-biomass spectra (except when all species have equal biomass), exhibit dome-shaped [130] and becomes less consistently right-skewed as one descends into finer taxonomic classifications [21,131], which are possibilities for size-biomass spectra that cannot be captured by power laws. At the extreme, ontogeny within many species leads to a greater total biomass for large adults than for small larvae (left skew) [21]. The possibilities of both left and right skews in addition to nonlinearity make standard distributions like lognormal, exponential, and gamma inappropriate because each only produces one type of skew. We used truncation because, without it, continuous distributions would typically imply finite biomass at unrealistic body sizes, especially for groups with high total biomass (e.g., bacteria having finite biomass at the size of trees). We also renormalized the distribution to retain the total biomass under the curve. Other similar distributions such as skew normal and extreme value can also be used, but they cannot be meaningfully distinguished from GEV because of the paucity of data, nor favored for mechanistic reasons because of a lack of theories on size-biomass relationships.

Two steps were involved in generating a bootstrapped estimate of median size-biomass spectra per group. We first interporate probability distributions (Eqs 1 and 2) to three observed reference sizes for each organismal group compiled from the literature: minimum, median, and maximum sizes (S1–S3 Tables). This fit was achieved by minimizing the sum of squares of the residuals between the three observed reference (log) sizes and the 0.05th, 50th, and 99.95th percentiles of the truncated generalized extreme value distribution. The probability distribution thus placed close to 99.9% of the biomass within the reported size range. Truncation was applied at two orders of magnitude below the reported minimum size, but not to the maximum size, to accommodate uncertainties associated with undetected small species and immature individuals. This assumption is compatible with empirical evidence across marine and terrestrial life with offspring being around two orders of magnitude smaller than adults in mass [132,133]. For microbes, offspring length ($L$) is around 0.2 to 0.5 times of the parent among model organisms [134]. Since volume (proportional to mass) is approximately 4/3 $\pi L^3$ [56], offspring mass is one to two orders of magnitude smaller than parent mass. We note that *Pseudomonas aeruginosa*, one of the best-known bacteria that live in a wide range of human and natural habitats, have offspring that are two orders of magnitude smaller than parents in mass [135]. The upper size limits are likely more accurate than the lower size limits because larger species are easier to observe; in addition, the upper limits are not influenced by

ontogeny, hence the asymmetry in truncation. We explored different truncation amounts to both lower and upper limits in sensitivity tests.

In the second step, we used the initial distribution fit from step one to represent our uncertainty in where the median biomass occurs within groups (S4 Fig). A probability distribution is by definition the uncertainty in a parameter's value; in this case the parameter is the median size because it is the most uncertain among the three datapoints that was fitted to data. We then resampled 1000 sets of these within-group median body size and biomass, keeping minimum and maximum sizes constant, and re-fit the truncated generalized extreme value distribution each time to generate bootstrapped size-biomass relationships. This way, even in cases where biomass estimates have low uncertainty, such as in grassland plants, uncertainty in median size leads to large uncertainty in biomass at each possible grass size. In particular, to propagate median size uncertainty, the median size was randomly generated from the initially fitted truncated generalized distribution per bootstrap. To propagate biomass uncertainty, we randomly sampled in log space using standard deviation $\sigma = \lambda/1.96$, where the fold uncertainty $\lambda$ correspond to the 95% confidence interval (with the log upper/lower bounds deviating by $\lambda$ from the log mean according to a lognormal error model) following previous report [30]. The 2.5th, 50th, and 97.5th percentiles of the bootstraps represent the lower bound, median, and upper bound of the within-group size-biomass spectra.

## Statistical trends and modes across groups

Global median size-biomass spectra and confidence intervals were obtained by cumulating biomass density (Gt biomass per log body size) of all groups in a habitat realm (or realms) centered at each size bin (1/40 of a log unit) per bootstrap. In other words, the cumulative biomass density is the biomass probability density and then normalized so that the area under the curve matches the total biomass within realm(s). In the main text, we simplified the term "biomass density" to "biomass." Statistical descriptions were obtained for three different classifications of organisms: all realms, terrestrial, and marine.

To fit statistical relationships between size and cumulative biomass in each habitat realm, we did not perform simple regressions directly on the best estimated spectra because 1) biomass datapoints are not independent across sizes within groups, and 2) the cross-taxa biomass totals in any size class depends on all groups in that size class, making the error structure correlated across the size range. To obtain confidence bounds, we relied on a parametric bootstrapped ensemble of possible size class–total biomass spectra (size-biomass spectra). For each bootstrap, the possible continuous size-biomass spectrum was sampled 40 times per log size class from -18 to 11 in the same way that it was plotted for visualization (size bin width was 1/40 of a log unit). We then performed statistical regressions on each of the 1000 bootstrap sampled sets. The 2.5th and 97.5th percentiles of the outputs at each size represented each regression model's 95% confidence bounds. The result is that the confidence bounds may not strictly resemble the regression models; for example, single Gaussian fits across bootstraps may identify different peaks and thus the upper and lower bounds across size may be multimodal (S4 Fig). Size bins with total biomass lower than $10^{-5}$ Gt (1000 t), which is an order of magnitude below the lower bound of amphibian biomass (the lowest among all groups), were not included as datapoints for the regression. A cutoff is necessary to avoid large or infinitely negative values after log transformation, which would prevent regression from proceeding.

We fit two kinds of regression models to test for trends in the amount of biomass across size classes across all taxa. For allometric power law relationships, ordinary least-squares regressions were performed to obtain power exponents $\beta$ that explain the discrete sampled log size-log biomass (x-y) relationships. For Gaussian mixture models, up to four modes

(components) were fit using an expectation maximization algorithm to minimize nonlinear least squares ('gauss1', 'gauss2', etc. in Matlab R2017a, MathWork, Natick, MA). During fitting for the Gaussian mixture, we added $\log_{10}(10^{-5})+1$ to log biomasses to ensure that the minimum value was 1; smaller values were already removed previously. For plotting, we subtracted $\log_{10}(10^{-5})+1$ from the solutions. We measured $R^2$ and the corrected Akaike Information Criterion (AICc) for model comparison [136], which results in means and standard deviations across bootstraps.

We additionally obtained power laws for two alternative types of size spectra using linear regressions (Table 1). First, the size-abundance spectra [137] replaces biomass with abundance. Abundance is biomass divided by body mass, so the power law exponent $\alpha$ for size (mass)-abundance is approximately the exponent for size (mass)-biomass minus one [3]. Second, the normalized size-biomass [8] replaces biomass with total biomass divided by the width of biomass size class, centered in the middle of the size class along the x-axis. In our data synthesis, the width is a constant of one in log size scale, since each point along the x-axis represents the biomass density, or biomass per log size unit. Consequently, normalized biomass $B_N$ at log size $x$ is $B_N = B/(10^{x+0.5}-10^{x-0.5})$ where $B$ is the cumulative biomass density at size $x$. By taking the log of both sides of this equation, we obtain $\log_{10}B_N = \log_{10}B-x-0.454$. Since $\log_{10}B-x$ is $\log_{10}(B/10^x)$, or $\log_{10}(\text{abundance})$, log normalized biomass in our data is just log abundance minus 0.454. Thus, the power law exponent for the normalized size-biomass spectrum is identical to $\alpha$.

## Sensitivity analyses

We repeat the regression analyses on global size-biomass spectra with datasets composed using different truncation limits for the within-group GEV distributions, different definitions of body size (ramets vs. genets), and different mass inclusivity (with vs. without metabolically inactive material) (S1 Fig, Table 5).

Changing truncation limits should affect the GEV distributional fit for within-group size-biomass spectra. In particular, we experimented with the different size truncation limits of [-1, +1] and [0,0] on log scale. A small-enough truncation window should result in a distribution that is relative flat like most continuous probability distributions that have at most one interior inflection point. This implies size-biomass distributions that approach uniform distributions. Additionally, a truncated uniform size-biomass distribution is expected to minimize biomass uncertainty propagation because all bootstraps will have the same size range and only variations from biomass uncertainty.

The unit 'genets' was dissolved into smaller units of ramets for the variant definition of body size. Grassland plants, seagrass, soil fungi, and hard corals were affected by the switch to the ramet definition (S1 Table). In particular, the original large size range for soil fungi was reduced but remained the largest among all groups. This large size range reflects the group's unique history of having evolved and lost multicellularity many times [138], and having indeterminate growth through hyphae [139] that manifest in all possible sizes up to the upper limits. Some of the referenced species exhibiting minimum, median, and maximum sizes were changed based on the alternative definition.

We re-calculated the biomass spectrum only including the portion of the world's biomass that is "metabolically active", which would exclude skeletons, wood, and subterranean microbes [140]. This affects both the body size and biomass of forest plants, grassland plants, mangroves, and hard corals (S2 Table). Excluding biomass with low metabolism potentially reduces all reported minimum, median, and maximum sizes we reference from the literature withing groups because this biomass is taken out of all genets or ramets (individuals). In all

cases we found that species with the minimum, median, and maximum sizes remained the same, but their sizes were reduced.

## Supporting information

**S1 Fig. Sensitivity of the global body size biomass spectrum to different assumptions.** Grey dotted curves are 95% confidence bounds from 200 resamples from within-group uncertainties. See Fig 1 for color reference and default assumptions. A. Same data as main text, except with truncations at 1 log g on either side of reported minimum and maximum sizes. B. Same data as main text, except with truncations at reported minimum and maximum sizes. C. Sizes are defined for ramets or clones instead of genets, with truncation at -2 log g below the reported minimum size. D. Mass with low metabolism is omitted from body size and biomass estimates (plant woody material, hard coral skeleton, and subterranean microbes), with truncation at -2 log g below the reported minimum size.
(PDF)

**S2 Fig. Regression analyses on abundance.** Data is the same as in main text, except biomass is replaced by abundance or normalized biomass (biomass divided by size class width). Rows represent habitat realms (A: All realms, B: Terrestrial, C: Marine). Grey curves represent 95% confidence intervals of the data, and blue curves represent 95% confidence intervals of the model from 1000 bootstraps. $\alpha$ is the mean power exponent, and ± indicate standard deviations across bootstraps. Regression results are identical whether it is performed on log abundance or log normalized biomass as the dependent variable, because the latter is only offset from the former by a constant (-0.454).
(PDF)

**S3 Fig. Group size range.** Size ranges of 36 groups are quantified as the log max:min size ratio, corresponding to the number of $\log_{10}$ units that each group spans in size (g). This quantity shows no relationship with median body size (on log-log scale), with a power exponent of 0.0 ±0.10 (S.D.) and a p-value of 0.99. The size ratio has a mean of 7.0±4.2.
(PDF)

**S4 Fig. Estimating within-group size-biomass spectrum.** The size-biomass relationship for each group is composed of biomass and size estimates. Biomass estimates and uncertainties were mostly based on published syntheses that incorporate multiple independent sets of sampled biomass (black dots on maps) that are projected over habitat ranges (akin to species distribution models). Body size distribution and uncertainty were based on literature search for minimum, median, and maximum sizes within groups (green dots). A truncated generalized extreme value distribution was first fitted to the three points that result in an uncertainty estimate for median size. 1000 pairs of resampled total biomass and median size were then used to refit a truncated generalized extreme value distribution, resulting in a set of bootstrap samples that create the final median estimate and 95% confidence intervals for the size-biomass spectrum.
(PDF)

**S1 Table. Body sizes measured for ramets instead of genets.**
(PDF)

**S2 Table. Body sizes excluding sizes and biomass with low metabolism.**
(PDF)

**S3 Table. Icon sources.** All icons belong to the public domain.
(PDF)

**S1 File. Supporting information references.**
(PDF)

## Author Contributions

**Conceptualization:** Eden W. Tekwa, Malin L. Pinsky.

**Data curation:** Eden W. Tekwa, Katrina A. Catalano, Anna L. Bazzicalupo.

**Formal analysis:** Eden W. Tekwa.

**Investigation:** Eden W. Tekwa.

**Methodology:** Eden W. Tekwa, Katrina A. Catalano, Anna L. Bazzicalupo, Mary I. O'Connor, Malin L. Pinsky.

**Validation:** Eden W. Tekwa.

**Visualization:** Eden W. Tekwa.

**Writing – original draft:** Eden W. Tekwa, Katrina A. Catalano, Anna L. Bazzicalupo.

**Writing – review & editing:** Eden W. Tekwa, Katrina A. Catalano, Anna L. Bazzicalupo, Mary I. O'Connor, Malin L. Pinsky.

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
