## [Decision Letter · Decision Letter 0]

10 Jan 2023

PONE-D-22-29135The Sizes of LifePLOS ONE

Dear Dr. Tekwa,

Thank you for submitting your manuscript to PLOS ONE and for your patience. I hope my two earlier updates justified the long review process. I have now secured three reviews of your manuscript. Let me remind you that the third reviewer is not submitting the review through the PLOS ONE platform. Rather this reviewer sent me their review via email. I have attached that third review in the form of a word document.   All three reviewers find the manuscript important and timely and recommend publication after revision. I concur with their assessment. Rather than issues that require new data, analyses, or a complete rewriting of the paper, the consensus is that the paper can be improved by clarifying numerous points and by striving to achieve a bit more perspective. Reviewer 3 thinks you overstate some of the claims in the paper. I strongly encourage you to submit a revised version of the manuscript that addresses the points raised during the review process. Since the audience of PLOS ONE is extremely diverse, I also encourage you to minimize jargon or clearly define all technical terms.  Once I see the revised version and the rebuttal statement, I can decide whether I can proceed to a final decision on the paper or seek a second round of reviews.

We look forward to receiving your revised manuscript.

Kind regards,

Hans G. Dam, Ph. D.

Academic Editor

PLOS ONE

Journal Requirements:

"The research was supported by the Tula Foundation; the Gordon and Betty Moore Foundation;

MITACS; the Coral Reef Alliance; and NSF awards OCE-1426891, DEB-1616821, and OISE-1743711."

"The research was supported by the Tula Foundation (ET); the Gordon and Betty Moore Foundation (MP); MITACS (ET); the Coral Reef Alliance (ET & MP); and NSF awards OCE-1426891, DEB-1616821, and OISE-1743711 (MP). The funders had no role in study design, data collection and analysis, decision to publish, or preparation of the manuscript."

Reviewers' comments:

Reviewer's Responses to Questions

**Comments to the Author**

1. Is the manuscript technically sound, and do the data support the conclusions?

Reviewer #1: Yes

Reviewer #2: Yes

2. Has the statistical analysis been performed appropriately and rigorously? 

Reviewer #1: I Don't Know

Reviewer #2: Yes

3. Have the authors made all data underlying the findings in their manuscript fully available?

Reviewer #1: Yes

Reviewer #2: No

4. Is the manuscript presented in an intelligible fashion and written in standard English?

Reviewer #1: Yes

Reviewer #2: No

5. Review Comments to the Author

Reviewer #1: The manuscript quantifies the body-mass biomass spectrum of all life on the globe, across marine and terrestrial habitats. This is an ambitious and novel endeavour (another recent work did a similar analysis but only for the marine habitats). The analysis is comprehensive and appear to be competently done (I am not a statistician and cannot comment on the details of the stats). The text is clearly written and the results are well placed in the theoretical context (though I have some minor comments). The results are well presented graphically. Overall, I find this a very inspiring manuscript.

I have reviewed this manuscript before and I only have minor comments:

Minor comments:

Line 6: I don’t understand this sentence. How can diverse organism types converge on a similar max and min mass? I understand it after having read the paper, but while reading the abstract I was confused.

Line 35. I don’t understand why 11 is cited here. While they discuss energy equivalence, the spectra they find has slopes close to zero than to -0.25.

Line 35-37: “reduced energy transfer”. Not all theories rely on energy transfer (e.g. my own K.H. Andersen and J.E. Beyer: Asymptotic size determines species abundance in the marine size spectrum. American Naturalist (168) 54-61 (2006), or Benoıt, E., & Rochet, M. J. (2004). A continuous model of biomass size spectra governed by predation and the effects of fishing on them. Journal of theoretical Biology, 226(1), 9-21.). I would say that the common denominator is predator-prey interactions governed by the rule that bigger organisms eat smaller organisms.

Figure 2: This figure can be made more readable by removing the superfluous axes labels on all the “internal” panels.

Line 221: What do you mean by “trophic structure ordered by size”? Is the size-distribution more regular in the marine realm?

Line 223. Competition for light is not the reason why primary producers in the ocean are small. It is competition for nutrients (smaller cells have a lower R* for diffusive uptake of dissolved matter).

Signed

Ken H Andersen

Reviewer #2: The manuscript by Tekwa and coauthors represents the first attempt to characterize the global size spectrum of the biomass of living organisms on Earth, based on previous published studies. The size of an organism provides a first order control on many aspects of physiology and ecology, and is increasingly used as an organizing quantity in models of life on Earth. The biomass of organisms of any given size range, i.e., the size distribution (or size spectrum) is thus an important quantity for a range of disciplines (biology, ecology, biogeochemistry, etc.). Yet very significant uncertainties exist around it. Recently, the study by Bar On et al., 2018, PNAS provided a first quantification of the biomass of all living organisms on Earth.

The manuscript provides an extension of this work, translating these biomasses into biomass size spectra, i.e., adding a size dimension to them. Thus this study is essentially based on a meta-analysis of published biomass and size range estimates for a variety of loosely defined taxonomic groups, and applies statistical approaches to derive size spectra from these quantities.

The Authors find that global size spectra for both marine and terrestrial organisms can be reasonably well described by power laws with a slope around -1, meaning that biomass is equally distributed across log-size intervals. This pattern applies better to marine rather than terrestrial organisms. This result more or less agrees with previous studies, and it provides a useful, more comprehensive confirmation. Significant deviations from a simple power law are also found, with higher biomasses relative to the power law at both the lowest and the highest sizes of the range, to the point that bi-modal distributions can also describe the data reasonably well. This may be related to the observation that diverse taxonomic groups share similar maximum and minimum sizes, where biomass tends to accumulate. These deviations are more marked for terrestrial organisms, which show a significant biomass accumulation near the largest sizes. Uncertainty is propagated throughout, which is commendable, and the Authors are the first to recognize that some of the results are tentative, given the large uncertainties remaining.

I found the results interesting and stimulating, and the methods applied are quite novel and likely will be useful for future, more detailed applications. I imagine this work could inspire future efforts to revise and strengthen the size spectra estimates, or apply them regionally, as new data become available.

For what I can judge — while I’m not an expert in the statistical methods applied — the approaches are reasonable and manage achieve a lot with relatively scarce data. Yes, there are plenty of approximations and simplifying assumptions, but they are generally clearly acknowledged, and build on previous studies. I also appreciated the emphasis the Authors put on propagating uncertainties in their reconstruction of biomass size spectra, and the valuable sensitivity analysis to choices such as size ranges, definition of individual organisms, and definition of biomass. I should add that the Authors did an excellent job by adding a comprehensive discussion of caveats and limitations related to their approach that addressed most of the doubts I had while reading the paper.

I have a few criticisms that are relatively minor and the Authors should be able to address in revision. The main suggestion is to try to improve the paper writing to increase readability, and make sure the Methods are as clear as possible. This may require some careful rewriting.

- Taxonomic divisions are not very consistent or meaningful, so it’s easy to over interpret some of the group-by-group patterns shown in the figures. E.g all fungi are lumped together, while plants or animals are separated into finer and finer subdivisions, although they may not be particularly meaningful ecologically or biologically. The rationale here seems to be the need to use previously published data, on which these taxonomic subdivisions are based. This may be fine although not very satisfying. That said, the Authors focus on global size spectra, so limitations in taxonomic subdivisions are not essential for the points made by the paper.

- Some choices of organism or biomass definition, or marine vs. terrestrial organisms are a bit questionable. E.g., I would only use organic C for the definition of biomass, and not include inorganic C as in coral skeletons. I would also classify mangroves as terrestrial rather than marine. Inclusion of structural biomass for trees is also somewhat questionable. That said, the sensitivity analysis addresses these points, and the Authors provide a fairly thorough discussion of these choices — which seem often based on previous work, and do not dramatically alter the main findings.

- Perhaps the main criticism is that, in terms of writing, the paper is often dense and hard to follow, and sometimes a bit sloppily written. This is particularly acute in some sections (see some specific comments in the following). I would encourage the Authors to do a thorough editing and some rewriting to strive to simplify and clarify the messages, keeping in mind that the readership may not be familiar with many of the concepts and techniques used.

- The abstract is at times confusing and could be substantially clarified, especially since the reader may not have read through the full paper yet. E.g., I wouldn’t know what lines 5-6 mean whiteout reading the full text; the ranges of sizes discussed should be clearly and quantitatively stated; uncertainties could be provided in the power-law exponents reported, as well as statistical significance of the fits; terms such as “small” and “large” (line 10) need to be contextualized, otherwise are meaningless; it s not clear what regression the R2 =0.86 (line 11) refers to; some detail could be added to the last sentence on the “bimodal” distribution, since up to that point power laws have been discussed, etc.

- Introduction, page 3: critical quantities, such as “size-biomass spectra”, “size-abundance spectra”, “normalized size-biomass spectra” could be better defined and introduced; right now they are a bit thrown haphazardly in the mix, and most readers are probably not familiar with the subtleties behind them. The Authors could even consider adding a “box” or table that clearly lays out the main quantities discussed in the paper, with relevant units, etc. I found myself often going back and forth between the introduction, results, and figures to remember what quantity corresponded to what.

- Terms such as “genet” and “ramet” that are not commonly used across disciplines should be better explained early on, ideally the first instance that they are used. While the idea of genet is discussed in some more detail (lines 316-322), that of ramet is not, and neither is clearly defined. I was not familiar with either terms and I would have appreciated a simple introduction to them.

- Fig. 1 —the main figure of the paper — is indeed very interesting, though at times difficult to read. The y axis is labeled as biomass, but I wonder if this is the integrated biomass in each log10 bin. I realize this is hard to pull off, but colors are at time confusing, and they definitely do not seem color-blind friendly. The taxonomic groups in the legend seem to be ordered by lowest to highest biomass, but this is not clearly stated. (I hope it’s the case, because it’s very useful to contextualize the results).

- Table 1 uses “exponential notation” e.g. 1.08E+1, which is not usually recommended and not very clear; powers of 10, or directly log10 values seem more appropriate. Looking at the note on the amphibian habitat rainforest, an area of 5.5011347 x 10^12 m2 is reported. The number of digits in this quantity is so large — are they all significant?

- Figure 3 should include the color legend, because otherwise the reader needs to jump back and fort between Fig. 1 and 3, and that makes it hard to interpret the figure.

- Figure 4 is not very exciting, and it’s hard to parse, with overlapping faint lines, text overlapping with lines, etc.

- I’m a bit confused by Figure 5 — in particular the gaussian model fits. E.g., looking at panels B/G/L, my understanding is that the model fits should be gaussian curves in this case, which plotted in log-log space should look more or less like inverted parabola; this doesn’t seem the case, but perhaps I’m missing something. Similarly, panels H/M should show 2 gaussian models, but there seem to be irregularities and “bumps” in the lines that should not be there in relatively smooth 2 gaussian mixtures, etc.

- Lines 202-203: I’m a bit skeptical that the same underlying constraint would apply to organism organizations as diverse as grasses and whales — this statement seem a bit speculative and not well supported.

- Line 264-267: I completely struggle with this sentence, please rephrase and clarify.

- Line 286 : “simple explanation” seems a bit of a straw man — what are the Authors referring to here?

- The Methods particularly could be revised to provide more clarity, especially related to the approach of deriving size spectra for specific groups based on published biomass and size ranges. In particular the description of the second step is a bit obscure and doesn’t have many details. I struggled through the section starting in line 369, especially lines 400-411. Some quantities are not even defined, e.g., “λ” (lines 408-409). The discussion of the truncation is also confusing; it also focuses mostly on the lower range, so it’s unclear if and what truncation was applied at the upper size range.

- The rationale for the use of truncated generalized extreme value distribution is not well explained. Furthermore, these are not commonly used distributions, so some introduction and detail could be helpful for a general reader (e.g. showing somewhere, e.g. in the SI, how they look like, as a function of typical parameter values, etc.). The Authors state that these distributions are used because they are flexible, but a better discussion of the appealing characteristics of these distribution could be included, besides their flexibility (many multi-parameter distributions are flexible, but not all may be appropriate to this problem). The idea of “truncation” in these distributions is likewise not very clear. Also, equation (1) uses both t(z) and t(x), but I think t(z) is a typo, since z is not defined anywhere.

- Related, the Authors could add some effort to convince the reader that, besides being flexible, the truncated generalized extreme value distributions actually apply reasonably well to represent biomass size spectra observations. Is there any study supporting that? Or is their applicability just theoretical, and based on their characteristics? I suspect the result would not dramatically change if other “reasonable” distributions were used, but perhaps that can be confirmed.

- Lines 378-379 : I’m not sure what the Authors mean with this sentence, please rephrase and clarify.

- Line 394-396, “with offspring being around two orders of magnitude smaller than adults” — this likely doesn’t apply to unicellular organisms.

- The Section “Statistical trends and modes across groups” feels a bit sloppily written and could benefit from some clarifications, especially the last paragraph. The derivation in lines 447 could be better laid out instead of just saying “with some arithmetic, we obtain, …” — maybe just show the main steps.

6. PLOS authors have the option to publish the peer review history of their article (what does this mean?). If published, this will include your full peer review and any attached files.

Reviewer #1: **Yes: **Ken H Andersen

Reviewer #2: No

---

## [Author Response · Author response to Decision Letter 0]

22 Feb 2023

PONE-D-22-29135

The Sizes of Life

PLOS ONE

Dear Dr. Tekwa,

Thank you for submitting your manuscript to PLOS ONE and for your patience. I hope my two earlier updates justified the long review process. I have now secured three reviews of your manuscript. Let me remind you that the third reviewer is not submitting the review through the PLOS ONE platform. Rather this reviewer sent me their review via email. I have attached that third review in the form of a word document. 

 All three reviewers find the manuscript important and timely and recommend publication after revision. I concur with their assessment. Rather than issues that require new data, analyses, or a complete rewriting of the paper, the consensus is that the paper can be improved by clarifying numerous points and by striving to achieve a bit more perspective. Reviewer 3 thinks you overstate some of the claims in the paper. I strongly encourage you to submit a revised version of the manuscript that addresses the points raised during the review process. Since the audience of PLOS ONE is extremely diverse, I also encourage you to minimize jargon or clearly define all technical terms. Once I see the revised version and the rebuttal statement, I can decide whether I can proceed to a final decision on the paper or seek a second round of reviews.

We look forward to receiving your revised manuscript.

Kind regards,

Hans G. Dam, Ph. D.

Academic Editor

PLOS ONE

Thank you for providing excellent suggestions for the revision. We have addressed all comments from the review in this version. We believe this version is much improved in terms of clarity and reference to the existing literature. We look forward to a decision on publication.

Reviewers' comments:

Reviewer's Responses to Questions

Comments to the Author

1. Is the manuscript technically sound, and do the data support the conclusions?

Reviewer #1: Yes

Reviewer #2: Yes

2. Has the statistical analysis been performed appropriately and rigorously? 

Reviewer #1: I Don't Know

Reviewer #2: Yes

3. Have the authors made all data underlying the findings in their manuscript fully available?

Reviewer #1: Yes

Reviewer #2: No

4. Is the manuscript presented in an intelligible fashion and written in standard English?

Reviewer #1: Yes

Reviewer #2: No

5. Review Comments to the Author

Reviewer #1: The manuscript quantifies the body-mass biomass spectrum of all life on the globe, across marine and terrestrial habitats. This is an ambitious and novel endeavour (another recent work did a similar analysis but only for the marine habitats). The analysis is comprehensive and appear to be competently done (I am not a statistician and cannot comment on the details of the stats). The text is clearly written and the results are well placed in the theoretical context (though I have some minor comments). The results are well presented graphically. Overall, I find this a very inspiring manuscript.

I have reviewed this manuscript before and I only have minor comments:

Minor comments:

Line 6: I don’t understand this sentence. How can diverse organism types converge on a similar max and min mass? I understand it after having read the paper, but while reading the abstract I was confused.

We have revised to state that many biological groups share similar body size ranges, and no single group dominates size ranges where cumulative biomass is highest (line 7).

Line 35. I don’t understand why 11 is cited here. While they discuss energy equivalence, the spectra they find has slopes close to zero than to -0.25.

Thanks for spotting the error. Reference 11 should be cited in the following sentence that reference an exponent of 0 (line 40).

Line 35-37: “reduced energy transfer”. Not all theories rely on energy transfer (e.g. my own K.H. Andersen and J.E. Beyer: Asymptotic size determines species abundance in the marine size spectrum. American Naturalist (168) 54-61 (2006), or Benoıt, E., & Rochet, M. J. (2004). A continuous model of biomass size spectra governed by predation and the effects of fishing on them. Journal of theoretical Biology, 226(1), 9-21.). I would say that the common denominator is predator-prey interactions governed by the rule that bigger organisms eat smaller organisms.

Thank you for the references, which are now included along with the explanation you proposed (line 37).

Figure 2: This figure can be made more readable by removing the superfluous axes labels on all the “internal” panels.

Figure edited as suggested.

Line 221: What do you mean by “trophic structure ordered by size”? Is the size-distribution more regular in the marine realm?

We meant that trophic position is determined by size (lines 226-227).

Line 223. Competition for light is not the reason why primary producers in the ocean are small. It is competition for nutrients (smaller cells have a lower R* for diffusive uptake of dissolved matter).

Corrected (line 229).

Signed

Ken H Andersen

Reviewer #2: The manuscript by Tekwa and coauthors represents the first attempt to characterize the global size spectrum of the biomass of living organisms on Earth, based on previous published studies. The size of an organism provides a first order control on many aspects of physiology and ecology, and is increasingly used as an organizing quantity in models of life on Earth. The biomass of organisms of any given size range, i.e., the size distribution (or size spectrum) is thus an important quantity for a range of disciplines (biology, ecology, biogeochemistry, etc.). Yet very significant uncertainties exist around it. Recently, the study by Bar On et al., 2018, PNAS provided a first quantification of the biomass of all living organisms on Earth.

The manuscript provides an extension of this work, translating these biomasses into biomass size spectra, i.e., adding a size dimension to them. Thus this study is essentially based on a meta-analysis of published biomass and size range estimates for a variety of loosely defined taxonomic groups, and applies statistical approaches to derive size spectra from these quantities.

The Authors find that global size spectra for both marine and terrestrial organisms can be reasonably well described by power laws with a slope around -1, meaning that biomass is equally distributed across log-size intervals. This pattern applies better to marine rather than terrestrial organisms. This result more or less agrees with previous studies, and it provides a useful, more comprehensive confirmation. Significant deviations from a simple power law are also found, with higher biomasses relative to the power law at both the lowest and the highest sizes of the range, to the point that bi-modal distributions can also describe the data reasonably well. This may be related to the observation that diverse taxonomic groups share similar maximum and minimum sizes, where biomass tends to accumulate. These deviations are more marked for terrestrial organisms, which show a significant biomass accumulation near the largest sizes. Uncertainty is propagated throughout, which is commendable, and the Authors are the first to recognize that some of the results are tentative, given the large uncertainties remaining.

I found the results interesting and stimulating, and the methods applied are quite novel and likely will be useful for future, more detailed applications. I imagine this work could inspire future efforts to revise and strengthen the size spectra estimates, or apply them regionally, as new data become available.

For what I can judge — while I’m not an expert in the statistical methods applied — the approaches are reasonable and manage achieve a lot with relatively scarce data. Yes, there are plenty of approximations and simplifying assumptions, but they are generally clearly acknowledged, and build on previous studies. I also appreciated the emphasis the Authors put on propagating uncertainties in their reconstruction of biomass size spectra, and the valuable sensitivity analysis to choices such as size ranges, definition of individual organisms, and definition of biomass. I should add that the Authors did an excellent job by adding a comprehensive discussion of caveats and limitations related to their approach that addressed most of the doubts I had while reading the paper.

I have a few criticisms that are relatively minor and the Authors should be able to address in revision. The main suggestion is to try to improve the paper writing to increase readability, and make sure the Methods are as clear as possible. This may require some careful rewriting.

- Taxonomic divisions are not very consistent or meaningful, so it’s easy to over interpret some of the group-by-group patterns shown in the figures. E.g all fungi are lumped together, while plants or animals are separated into finer and finer subdivisions, although they may not be particularly meaningful ecologically or biologically. The rationale here seems to be the need to use previously published data, on which these taxonomic subdivisions are based. This may be fine although not very satisfying. That said, the Authors focus on global size spectra, so limitations in taxonomic subdivisions are not essential for the points made by the paper.

We have added lines 72-74 to incorporate the points you made here.

- Some choices of organism or biomass definition, or marine vs. terrestrial organisms are a bit questionable. E.g., I would only use organic C for the definition of biomass, and not include inorganic C as in coral skeletons. I would also classify mangroves as terrestrial rather than marine. Inclusion of structural biomass for trees is also somewhat questionable. That said, the sensitivity analysis addresses these points, and the Authors provide a fairly thorough discussion of these choices — which seem often based on previous work, and do not dramatically alter the main findings.

For mangroves in particular, it is definitely debatable, but we have added rationales and references for placing them in the marine realm (lines 321-322).

- Perhaps the main criticism is that, in terms of writing, the paper is often dense and hard to follow, and sometimes a bit sloppily written. This is particularly acute in some sections (see some specific comments in the following). I would encourage the Authors to do a thorough editing and some rewriting to strive to simplify and clarify the messages, keeping in mind that the readership may not be familiar with many of the concepts and techniques used.

Thank you for helping improve the clarity of the manuscript. We respond in particular to the specific points below.

- The abstract is at times confusing and could be substantially clarified, especially since the reader may not have read through the full paper yet. E.g., I wouldn’t know what lines 5-6 mean whiteout reading the full text; the ranges of sizes discussed should be clearly and quantitatively stated; uncertainties could be provided in the power-law exponents reported, as well as statistical significance of the fits; terms such as “small” and “large” (line 10) need to be contextualized, otherwise are meaningless; it s not clear what regression the R2 =0.86 (line 11) refers to; some detail could be added to the last sentence on the “bimodal” distribution, since up to that point power laws have been discussed, etc.

Lines 5-6 are revised to clarify that many biological groups align in body size ranges. We have added statistics on size and biomass. The R2=0.86 figure refers to the Gaussian mixture model, and we have moved the figure to earlier in the sentence to improve clarity (line 13). At the beginning of line 12, “bimodal” is added.

- Introduction, page 3: critical quantities, such as “size-biomass spectra”, “size-abundance spectra”, “normalized size-biomass spectra” could be better defined and introduced; right now they are a bit thrown haphazardly in the mix, and most readers are probably not familiar with the subtleties behind them. The Authors could even consider adding a “box” or table that clearly lays out the main quantities discussed in the paper, with relevant units, etc. I found myself often going back and forth between the introduction, results, and figures to remember what quantity corresponded to what.

Thank you for this suggestion. We have added a new Table 1 defining the key terms.

- Terms such as “genet” and “ramet” that are not commonly used across disciplines should be better explained early on, ideally the first instance that they are used. While the idea of genet is discussed in some more detail (lines 316-322), that of ramet is not, and neither is clearly defined. I was not familiar with either terms and I would have appreciated a simple introduction to them.

We have added definitions for genet and ramet when they first appear (lines 148-149) and explained these better in the Methods (lines 34-336).

- Fig. 1 —the main figure of the paper — is indeed very interesting, though at times difficult to read. The y axis is labeled as biomass, but I wonder if this is the integrated biomass in each log10 bin. I realize this is hard to pull off, but colors are at time confusing, and they definitely do not seem color-blind friendly. The taxonomic groups in the legend seem to be ordered by lowest to highest biomass, but this is not clearly stated. (I hope it’s the case, because it’s very useful to contextualize the results).

The y-axis represents the cumulative biomass per log size centered at the location along the x-axis. The size bins are not separate but overlap because each biomass measure is centered with a rolling window, if that makes sense. Technically the y-axis represents the cumulative biomass density, which is the biomass probability density scaled up so that the area under the curve matches the global biomass. This is now better explained (lines 451-455).

Apologies for the color, but we were limited for choices because of the large number of biological groups. The colours were picked so that on a grey scale they are discernable from adjacent groups, and icons were used in conjunction with the legend to ensure that the group names can be identified for color-blind accessibility. The caption is now improved to clarify how to read the cumulative and group biomass, and how groups are ordered.

- Table 1 uses “exponential notation” e.g. 1.08E+1, which is not usually recommended and not very clear; powers of 10, or directly log10 values seem more appropriate. Looking at the note on the amphibian habitat rainforest, an area of 5.5011347 x 10^12 m2 is reported. The number of digits in this quantity is so large — are they all significant?

We have reduced the number of digits and changed the scientific notation to x10#.

- Figure 3 should include the color legend, because otherwise the reader needs to jump back and fort between Fig. 1 and 3, and that makes it hard to interpret the figure.

The color legend is added to Figure 3.

- Figure 4 is not very exciting, and it’s hard to parse, with overlapping faint lines, text overlapping with lines, etc.

We have moved this to the supplementary Figure S2 (and rearranged supplementary figures in sequence of appearance in main text).

- I’m a bit confused by Figure 5 — in particular the gaussian model fits. E.g., looking at panels B/G/L, my understanding is that the model fits should be gaussian curves in this case, which plotted in log-log space should look more or less like inverted parabola; this doesn’t seem the case, but perhaps I’m missing something. Similarly, panels H/M should show 2 gaussian models, but there seem to be irregularities and “bumps” in the lines that should not be there in relatively smooth 2 gaussian mixtures, etc.

The confidence intervals are from bootstrapped gaussian regressions. For Gaussian 1 (one component), while the fit to each bootstrap (and to the original data) is Gaussian (normal), regressions across multiple bootstraps will result in upper and lower bounds at each body size that together not necessarily be normal. This is now explained on lines 466-468. This was also illustrated in Figure S4.

- Lines 202-203: I’m a bit skeptical that the same underlying constraint would apply to organism organizations as diverse as grasses and whales — this statement seem a bit speculative and not well supported.

We have added the possibility of multiple mechanisms to line 207.

- Line 264-267: I completely struggle with this sentence, please rephrase and clarify.

Agreed. See revised lines 300-302.

- Line 286 : “simple explanation” seems a bit of a straw man — what are the Authors referring to here?

Revised to “existing explanations” (line 302).

- The Methods particularly could be revised to provide more clarity, especially related to the approach of deriving size spectra for specific groups based on published biomass and size ranges. In particular the description of the second step is a bit obscure and doesn’t have many details. I struggled through the section starting in line 369, especially lines 400-411. Some quantities are not even defined, e.g., “λ” (lines 408-409). The discussion of the truncation is also confusing; it also focuses mostly on the lower range, so it’s unclear if and what truncation was applied at the upper size range.

λ is the fold uncertainty as reported by Bar-on 2018, now clarified on line 445. In the main analysis, truncation was not applied to the maximum size (line 424) for reasons given on lines 431-432, but different truncations on either side were tested in sensitivity analyses. The second step of using the initial probability distribution fit as the median size uncertainty is now better explained (lines 4435-438), appealing to the definition of probability distribution and our focus on the median as the uncertain observation.

- The rationale for the use of truncated generalized extreme value distribution is not well explained. Furthermore, these are not commonly used distributions, so some introduction and detail could be helpful for a general reader (e.g. showing somewhere, e.g. in the SI, how they look like, as a function of typical parameter values, etc.). The Authors state that these distributions are used because they are flexible, but a better discussion of the appealing characteristics of these distribution could be included, besides their flexibility (many multi-parameter distributions are flexible, but not all may be appropriate to this problem). The idea of “truncation” in these distributions is likewise not very clear. Also, equation (1) uses both t(z) and t(x), but I think t(z) is a typo, since z is not defined anywhere.

The reason for using a flexible model is that the literature has observed both left and right skews for similar data. Because of the lack of data and theory on size-biomass relationships, there is no way to determine whether different distributions that also exhibit left and right-skews are better fits. We just have to pick one, so we chose GEV. This is now better explained on lines 386-393. The issue of truncation is now discussed more elaborately on lines 414-416. The main point is that truncation is needed to avoid unrealistic scenarios such as having some bacteria the size of trees; otherwise most continuous distributions have finite probabilities of large size deviations. Examples of the GEV left and right-skew distributions are incorporated into Figure S4 and is now referenced on line 393.

- Related, the Authors could add some effort to convince the reader that, besides being flexible, the truncated generalized extreme value distributions actually apply reasonably well to represent biomass size spectra observations. Is there any study supporting that? Or is their applicability just theoretical, and based on their characteristics? I suspect the result would not dramatically change if other “reasonable” distributions were used, but perhaps that can be confirmed.

GEV can describe nonlinear relationships (on log-log scale) that appear likely empirically, whereas power laws cannot (lines 389-392). A recent study of plant communities (Dillon et al. 2019), though not globally extrapolated, used a Weibull distribution, which is a special case of GEV (line 388). Since there are only three reliable datapoints per biological group, it is not biologically meaningful to distinguish between distributions similar to GEV that can exhibit both left and right skews. This is pointed out on lines 399-423. The results (Figure 2) are reasonable as far as we can discern and compare well to current knowledge for comparable studies on fish and plants (lines 114-115).

- Lines 378-379 : I’m not sure what the Authors mean with this sentence, please rephrase and clarify.

The sentence does not seem necessary so we have taken it out.

- Line 394-396, “with offspring being around two orders of magnitude smaller than adults” — this likely doesn’t apply to unicellular organisms.

We have added citations on known microbial parent-offspring mass ratio, which range from 1 to 2 orders of magnitude. We note that Pseudomonas aeruginosa, a common bacteria found in human and natural habitats, exhibits a ratio at 2 orders of magnitude (lines 427-431).

- The Section “Statistical trends and modes across groups” feels a bit sloppily written and could benefit from some clarifications, especially the last paragraph. The derivation in lines 447 could be better laid out instead of just saying “with some arithmetic, we obtain, …” — maybe just show the main steps.

We have revised the section and rewritten the last paragraph to clarify how the different spectra are constructed (lines 483-488). On line 491, we now explicitly state that the arithmetic is taking the log of both sides of the equation on line 490.

Reviewer #3:

Overall, I think the study should be worthy of publication and has the potential to be highly regarded and widely cited. Unfortunately, the way it is written does not endear the informed reader to the study. I think it oversells its novelty, fails to cite key prior work in some relevant places and omits discussion of its limitations and assumptions. I think these items are straightforward to address if the authors are willing and assuming these items can be fixed, I would recommend publication.

The paper claims that it remains unclear how biomass is distributed across body sizes (e.g. first sentence of abstract), and yet this has been extensively studied for over 50 years in aquatic systems and so is really only true in terrestrial systems. I think the paper should reserve its claims of novelty to the terrestrial realm and be more honest in its largely confirmatory status for aquatic systems. The paper needs to better delineate its actual contribution while acknowledging the extensive prior work in this field. Personally, I think the introductory remarks about energetics and size-density scaling at the species level could be replaced or expanded with a better treatment of the size spectrum in aquatic systems, concluding that very little has been done in terrestrial systems, and that this work fills that gap by confirming the aquatic results, and extending size distributions to terrestrial and truly global realms, albeit with extensive assumptions.

We agree with clarifying the novelty and bringing prior marine spectrum works to the forefront (abstract line 11, line 29). Perhaps the key is stating that the novel contribution is compiling size-biomass data of all living things (line 3), which contrasts from previous efforts, including aquatic studies, that were relatively comprehensive yet incomplete.

Size spectra for the marine realm has been extensively studied, yet they invariably did not include marine organisms with high biomass (corals, seagrass, macroalgae, mangroves). These make up 45% of the marine biomass according to our estimates (lines 119-121, 218-219). Even if corals and mangroves’ inclusions are debatable (but see lines 321-322 for additional rationale), seagrass and macroalgae are certainly important marine components at the large size range. Even though some of the resulting patterns are similar to previous marine studies, the data and approach are quite different, so it’s reasonable to interpret our study not as aiming to be confirmatory, only that the results are partly confirmatory. The resulting marine patterns, which are more complete but more uncertain than previously documented, do not take away the scope of the novelty we emphasized (more comprehensive biological group inclusion, body size data, error propagation, and testing different biological and statistical assumptions). For these reasons, we think it is accurate to state that size-biomass spectra were previously unclear for all realms.

The paper has over 100 citations, but is light on highlighting prior work on the size spectrum. The paper also does not cite Bar-On et al. 2019 (its primary source of biomass data) or Hatton et al. 2021 (a prior global synthesis of marine systems) appropriately, IMO. Several sentence are also missing citations to prior work, despite explicit mention of such prior work (e.g. line 189, 227, 275, 278). Overall, the work would be better received by aquatic ecologists if the work can better integrate itself into the size spectrum literature.

We agree that additional citations are important, and we have added new and existing citations on the lines mentioned (now lines 191, 229, 274, 293).

Finally, although I was pleased with some of the supplementary analysis investigating the sensitivity of assumptions to results, it was difficult to evaluate in the context of how these assumptions might impact the overall findings. I think there should be a dedicated sub-section of the discussion that is more open about the limitations of the study. For example, how much do the results depend on the results of Bar-On et al., or the assumptions about within group size distributions?

We have added to the paragraph on uncertainties in the Discussion, highlighting that biomass and uncertainty estimates partly rely on Bar-On’s work but were supplemented and checked against newer estimates where available (lines 266-267). We clarified that the distribution we used for within-group size-biomass relationships avoids the stronger assumptions that would be made by alternatives (lines 280-282).

I believe with a general toning down of the claims of novelty, a better treatment of prior work and a brief acknowledgements of the limitations, this paper would be acceptable and will be a valuable study. Below are listed a number of more specific suggestions that complement the three above-mentioned general areas for improvement.

Line 29: I think it is misleading to claim that the theories to explain Damuth’s law, or metabolic scaling have been “successful” – they continue to be debated and it seems safer to describe them as possible explanations.

Agree. Replaced “successful” with “applied” (line 30).

Line 32: Size-density scaling of separate species with exponent ¾ is sometimes observed, and often not. If this is deemed relevant to the findings of this paper, it should be made clear that this is far from universal (and possibly due to underestimates on the slope due to OLS with ~3 orders of magnitude residual variation).

Updated to highlight the large residuals within groups and deviations between groups, with a new citation (Hatton et al. 2019 PNAS) (line 37).

Line 35: I don’t believe it is well established that the size-spectrum slope of -1 is due to reduced energy transfer across trophic levels, as this sentence suggests. It may be reasonable, but not well established.

As another reviewer suggested, size-ordered predator-prey interactions are better explanations than energy transfer for the exponent of -1, and we noted that some hypotheses also predict exponents less than -1 (line 37).

Line 46: The paper by Hatton et al. 2021 (Ref 21) is not properly cited. It is first cited in support of the dominance of medium sized plankton (which is incorrect), and later suggestive that they made misleading claims about the scope of their work. This study undertook analysis of the global marine size spectrum and should be cited appropriately, both acknowledging its precedence for the marine realm in the introduction and its results at line 117 and line 212.

We did misread Hatton’s conclusion on the marine size spectrum. We drew the observation of medium size plankton dominating from their Figure 3A, but this figure plots biomass by biological group and not strictly by size. Their size-biomass spectrum shows that microbes dominate. This is now corrected on line 52. Even though the study was the most comprehensive marine spectrum to date, it did leave out major marine habitat builder groups (comprising 45% of marine biomass). We think noting commonly missing taxa in multiple “bacteria-to-whale” studies is not a particular criticism of Hatton’s comprehensive study, just that it is a common and limited convention on what we think is marine life. We have now cited the study earlier in the introduction when introducing aquatic spectra (line 29).

Line 97: Nearly all biomass data are from Bar-On et al. 2019, and so this should be stated up-front in the introduction, since results should heavily depend on this single reference. It thus also does not appear to be cited appropriately until the Methods, which comes across as misleading. For example at line 72 (“The resulting catalogue of biomass data matched to body sizes stands as a record of present knowledge about life on Earth”), it seems to be misleading not to cite the source for the biomass data. It is also the primary source for one of the three major conclusions.

Agreed, now cited on line 74 in the Introduction.

Line 190: The discussion lists three major biological findings, the first of which I did not understand (unless it is the trivial finding that different taxonomic groups can have similar body sizes), the second of which is confirmatory of prior work going back to Sheldon et al. 1972 (at least in aquatic systems), and the third of which is a restatement of prior work such as Bar-On et al. 2019. These and other references to prior work need to be explicitly acknowledged.

Agreed that the first conclusion is unclear; we have added “these extreme sizes contain most of the biomass.” The second is not strictly a confirmation of previous studies because Sheldon targeted a limited range of species in a community, in contrast to our global study and require different explanations. We think the similar exponents require more nuanced discussions (lines 219-222). The third conclusion cannot be drawn directly from Bar-On’s paper, and they did not claim the same. Their paper showed that there is more biomass on land than in the ocean. In contrast, our third conclusion states that on land, more biomass is concentrated in large organisms than in smaller organisms, which differs from the ocean. This pattern can only be shown with body size data. All relevant papers including Sheldon and Bar-On are cited throughout the Discussion.

The paragraph starting at line 197 makes some poorly justified claims, in my opinion. On one hand, the range of 10^7 to 10^9 (two decades) is considered a narrow range, while on the other hand, further down the paragraph, the roughly one order of magnitude greater biomass at the extremes of the size spectrum is considered significant (despite the considerably greater uncertainty in these estimates). Both claims (coincidence in max body sizes and significant internal modes) appear quite weak in my opinion.

We stated that the one order magnitude greater biomass is less than the uncertainty magnitude, so we did not state that such a peak is undisputable and indeed highlighted that such a peak is uncertain. In contrast, the two-order size range that constitutes the upper range where multiple groups max out at is narrow relative to the 26 orders of magnitude in size for all living things. We do not think these statements are in conflict with each other.

The paragraph starting at line 220 makes a cursory attempt to explain the differences between terrestrial and aquatic size spectra. Despite the introduction of the paper and this paragraph itself making mention of the energetic considerations that have dominated this literature, there is no mention of the presence of woody, non-metabolically active material. It is excellent that the authors undertook this analysis (Fig. S3 D) and I think it should be mentioned as a contributing factor to the hump, though possibly not sufficient to account for it. This gives more credence to a possible energetic basis for the origin of the size spectrum. At least I would expect the authors to make mention of woody material in this paragraph (in addition to line 255), and cite their SM analysis.

We have now revised the paragraph to discuss the sensitivity analysis on metabolically inactive material and the statistical power law changes (lines 235-239).

Methods

Perhaps I am not understanding the methods, but fitting three points (min, median and max) to a three parameter distribution does not inspire confidence, especially when these input data are so uncertain and each group somewhat arbitrary. I am thus not convinced that any hump in the distribution is significant, and so don’t see a strong justification that non-linear methods should be superior to a linear fit on log-axes (line 270).

If we simply used the generalized extreme value (GEV) fit to the data as our final size-biomass estimates within groups, then we would also have no confidence because of the lack of data. However, this is not what we have done. Instead, we used bootstrapped regressions, over which a 3-parameter model interpolated to 3 datapoints provides sufficient flexibility to accurately capture the expected distribution and its uncertainty in each group (it can be left or right skewed, symmetrical, uniform, or power law-like). This is because bootstrapped GEV avoids strong assumptions about the shape (median, spread, skew) and similarity of the empirical distributions across groups (lines 520-526), assumptions that would violate variations in distributional shapes previously observed in related spectral studies at different taxonomic levels (lines 398-416).

Line 325: What is the reference for “assuming a log-normal size-biomass distribution”?

The reference appearing in the middle of the sentence should be near the end (line 387), which contains this assumption.

Line 330: Assuming sizes at maturity are representative of a group are particularly problematic for fish and trees, where “maturity” may be several orders of magnitude larger than juveniles and could skew results noticeably (even on log scales).

The literature on size distribution within species, such as the reference on line 349 as well as line 408 (#19: Andersen et al. 2016 on fish), shows that most of the biomass should be concentrated near the maximum size despite potentially higher abundance of juveniles. Given no other general empirical knowledge across groups, the assumption that a reported species’ mature size is representative of the group’s minimum, median, or maximum sizes is to our knowledge the best-informed choice.

Line 365: I did not understand the sentence beginning at this line.

We have rewritten the paragraph (lines 377-390) to better explain the rationale for testing the relationship between size and size range. This resolves the confusions caused by the cited line.

Line 457: I did not understand this paragraph.

We have expanded the paragraph to explain how truncation limits affect the fit and uncertainty propagation (lines 520-526).

Table S2: I did not understand this table. I think this is a valuable analysis and so some further description is warranted.

This sensitivity analysis was described on lines 534-537. We added lines 537-540 to explain why excluding metabolically inactive biomass reduces the reported body sizes that we reference.

I wish the authors the best of luck in addressing my concerns.

---

## [Editor Report · Decision Letter 1]

1 Mar 2023

The Sizes of Life

PONE-D-22-29135R1

Dear Dr. Tekwa,

We’re pleased to inform you that your manuscript has been judged scientifically suitable for publication and will be formally accepted for publication once it meets all outstanding technical requirements.

Kind regards,

Hans G. Dam, Ph. D.

Academic Editor

PLOS ONE

Additional Editor Comments (optional):

Dear Dr. Tewka:

I am satisfied that you have addressed the reviewers' concerns and suggestions.

There is an unintended error on line 27 that needs to be removed from your final document file: "see Error! Reference source not found."
---

## [Editor Report · Acceptance letter]

7 Mar 2023

PONE-D-22-29135R1 

The sizes of life 

Dear Dr. Tekwa:

I'm pleased to inform you that your manuscript has been deemed suitable for publication in PLOS ONE. Congratulations! Your manuscript is now with our production department. 

Kind regards, 

on behalf of

Dr. Hans G. Dam 

Academic Editor

PLOS ONE